# A flexible framework for simulating and fitting generalized drift-diffusion models

Maxwell Shinn[1,2†], Norman H Lam[3†‡], John D Murray[1,2,3*]

[1]Department of Psychiatry, Yale University, New Haven, United States; [2]Interdepartmental Neuroscience Program, Yale University, New Haven, United States; [3]Department of Physics, Yale University, New Haven, United States

**Abstract** The drift-diffusion model (DDM) is an important decision-making model in cognitive neuroscience. However, innovations in model form have been limited by methodological challenges. Here, we introduce the generalized drift-diffusion model (GDDM) framework for building and fitting DDM extensions, and provide a software package which implements the framework. The GDDM framework augments traditional DDM parameters through arbitrary user-defined functions. Models are solved numerically by directly solving the Fokker-Planck equation using efficient numerical methods, yielding a 100-fold or greater speedup over standard methodology. This speed allows GDDMs to be fit to data using maximum likelihood on the full response time (RT) distribution. We demonstrate fitting of GDDMs within our framework to both animal and human datasets from perceptual decision-making tasks, with better accuracy and fewer parameters than several DDMs implemented using the latest methodology, to test hypothesized decision-making mechanisms. Overall, our framework will allow for decision-making model innovation and novel experimental designs.

**\*For correspondence:**
john.murray@yale.edu

[†]These authors contributed equally to this work

**Present address:** [‡]Department of Brain and Cognitive Sciences, Massachusetts Institute of Technology, Cambridge, United States

**Competing interests:** The authors declare that no competing interests exist.

## Introduction

The drift-diffusion model (DDM) is an important model in cognitive psychology and cognitive neuroscience, and is fundamental to our understanding of decision-making (*Ratcliff, 1978*; *Bogacz et al., 2006*). The DDM explains both choice and response time (RT) behavior across a wide range of tasks and species, and has proven to be an essential tool for studying the neural mechanisms of decision-making (*Gold and Shadlen, 2007*; *Forstmann et al., 2016*; *Ratcliff et al., 2016*). As a sequential sampling model for two-alternative forced-choice tasks, it assumes the subject obtains a continuous stream of evidence for each alternative throughout the course of an experimental trial. Evidence consists of some true underlying signal (drift) in addition to neural and environmental noise (diffusion). According to the DDM, once a decision variable tracking the difference in evidence between the two alternatives is sufficiently large, i.e. when the decision variable reaches some fixed upper or lower bound, a choice is made at the moment of bound crossing, thereby modeling for the decision both choice and RT. In its simplest form, the DDM includes four parameters: a drift rate, representing evidence; a diffusion constant or bound height representing noise or response caution; a starting position, representing side bias and often fixed at zero; and a non-decision time, representing afferent and efferent delays but external to the DDM process (*Ratcliff, 1978*). These models can be fit to choice and response time data.

In order to test new hypotheses about decision-making processes, there is great interest in developing extensions to the DDM. For example, the three-parameter DDM is known to be sub-optimal with respect to maximizing reward rate in standard block-design tasks when evidence strength varies unpredictably (*Malhotra et al., 2018*; *Evans and Hawkins, 2019*). Therefore, it has been hypothesized that decision-making is governed by an urgency signal which increases the probability of response over the course of a trial, implemented as either time-varying evidence gain (*Cisek et al.,*

*2009*; *Thura et al., 2012*; *Murphy et al., 2016*; *Ditterich, 2006b*; *Drugowitsch et al., 2014a*) or collapsing integration bounds (*Churchland et al., 2008*; *Hawkins et al., 2015a*; *Ditterich, 2006b*; *Drugowitsch et al., 2012*). Additionally, the DDM may be extended in order to model more diverse experimental paradigms. For example, evidence which changes in magnitude throughout a single trial violates the DDM's assumption that drift rate is fixed for a single trial. Modeling such tasks requires DDM extensions to support time-varying drift rate. We will show that these and most other extensions can be encapsulated within a single generalized DDM (GDDM). The GDDM provides a common mathematical language for discussing DDM extensions, and as we will show, supports an efficient framework for simulating models and fitting them to data.

There are three main methods for obtaining response time distributions from the GDDM. First, analytical solutions have a fast execution time, but only exist for special cases of the GDDM. While efficient, most GDDMs do not permit analytical solutions, and thus analytical solutions are only common for the DDM. Several packages exist for analytically solving DDMs and a limited number of extensions (*Wiecki et al., 2013*; *Wagenmakers et al., 2007*; *Grasman et al., 2009*; *Vandekerckhove and Tuerlinckx, 2008*; *Voss and Voss, 2007*; *Drugowitsch, 2016*; *Tavares et al., 2017*; *Millner et al., 2018*; *Srivastava et al., 2017*). A second method, trial-wise trajectory simulation, fully supports GDDMs for flexibility in model design (*Chandrasekaran and Hawkins, 2019*). However, this method is inefficient, and shows limited precision and computational efficiency. We will show that this in practice impacts the ability to fit data to RT distributions. To our knowledge, the only software package which currently exists for simulating GDDMs uses trial-wise trajectory simulation (*Chandrasekaran and Hawkins, 2019*).

A third and better method is to iteratively propagate the distribution forward in time, which is achieved by solving the Fokker-Planck equation (*Voss and Voss, 2008*) or using the method of *Smith, 2000*. These methods allow the RT distribution of GDDMs to be estimated with better performance and lower error than trial-wise trajectory simulation. However, they are difficult to implement, which has thus far impeded their widespread use in cognitive psychology and neuroscience. Currently, GDDMs without known analytical solutions are best implemented using home-grown code (e.g. *Ditterich, 2006a*; *Zylberberg et al., 2016*; *Hawkins et al., 2015b*; *Brunton et al., 2013*), which may not utilize the latest methodology or, as emphasized by *Ratcliff and Tuerlinckx, 2002*, could yield misleading results. Progress is limited by the ability to simulate and fit extensions to the DDM. A common open-source software package for efficiently solving GDDMs would help promote code sharing and reuse, and reduce the amount of time which must be spent implementing diffusion models. Researchers need a general, highly-efficient toolbox for building, solving, and fitting extended DDMs.

Here, we formally define and evaluate the GDDM framework for solving and fitting extensions to the DDM, and introduce the PyDDM (https://github.com/murraylab/PyDDM) software package as an implementation of this framework. We show that a GDDM with scientifically-interesting cognitive mechanisms can capture experimental data from a perceptual decision-making task in an animal dataset better than alternative models, while still providing an excellent fit to human data. Then, we show the execution time vs error tradeoff induced by different numerical methods, and demonstrate the GDDM RT distributions may be estimated with short runtime and low error. In particular, GDDMs may utilize the Crank-Nicolson and backward Euler methods to numerically solve the Fokker-Planck equation, which we will show to be highly efficient compared to alternative methods, and which permits fitting with full-distribution maximum likelihood. Solving the Fokker-Planck equation is analogous to simulating an infinite number of individual trials. The GDDM together with its associated fitting methodology comprise the GDDM framework. Finally, we compare our implementation of the GDDM framework to other software packages for diffusion modeling. In addition to being the only package to support the GDDM framework, we show PyDDM has a number of technical advantages, such as near-perfect parallel efficiency, a graphical user interface for exploring model forms, and a software verification system for ensuring reliable results. We hope that the GDDM framework will encourage innovative experimental designs and lower the barrier for experimenting with new model mechanisms.

## Results

### GDDM as a generalization of the DDM

The classic DDM is a four-parameter model which specifies fixed values for drift, bound height or noise level, starting position bias, and non-decision time (*Figure 1*). It assumes that evidence is constant throughout the trial, and that the integration process does not directly depend on time. This form is mathematically accessible, and has been used to model choice and RT data (*Ratcliff, 1978*; *Ratcliff et al., 2016*).

In order to expand our knowledge of decision-making mechanisms, there is interest in using the DDM to model a variety of experimental paradigms. To accommodate different paradigms, the DDM itself must be extended. One key example is the case where a choice bias is induced experimentally via a change in prior probability or reward magnitude, and it is hypothesized that an offset starting position alone is insufficient to capture the behavioral effects. The DDM has been extended to accommodate an experimentally-induced bias by placing a constant offset on the drift rate (*Mulder et al., 2012*; *Ratcliff, 1985*; *Ashby, 1983*).

There is a set of influential yet insufficient extensions to the DDM called the 'full DDM'. This set of extensions was developed to explain two specific discrepancies between the DDM and data (*Anderson, 1960*; *Laming, 1968*; *Blurton et al., 2017*). First, experimental data exhibited a difference in mean RT between correct and error trials which could not be captured by the classic DDM, so two parameters for across-trial variability were introduced to explain this difference: uniformly-distributed initial conditions to explain fast errors (*Laming, 1968*) and Gaussian-distributed drift rate to explain slow errors (*Ratcliff, 1978*; *Ratcliff and Rouder, 1998*). Second, the classic DDM also had a sharper rise time than experimental data, so uniformly-distributed across-trial variability in non-decision time was introduced to capture this effect. A previous study validated predictions of these across-trial variability parameters (*Wagenmakers et al., 2008a*). Compared to the classic DDM, the 'full DDM' improves the fit to data, but does not expand the range of experiments which may be performed. Likewise, it does not facilitate extensions to test alternative cognitive mechansims.. A more versatile framework is therefore needed.

The GDDM framework provides a flexible means for developing extensions to the DDM (*Figure 1*) which can accommodate a wide range of experimental designs and hypotheses about decision-making. The GDDM allows for extensions in five key aspects. First, the drift rate and noise level may depend on time. This allows a range of new experimental paradigms to be modeled using the DDM, such as doubly-stochastic evidence (*Zylberberg et al., 2016*), pulsed evidence (*Huk and Shadlen, 2005*), discrete evidence (*Cisek et al., 2009*; *Brunton et al., 2013*; *Pinto et al., 2018*), or changing evidence (*Shinn et al., 2020*). Additionally, the time dependence of drift rate and noise allows a time-variant urgency signal which modulates the gain of evidence (*Cisek et al., 2009*; *Thura et al., 2012*; *Murphy et al., 2016*; *Drugowitsch et al., 2014a*). Second, drift rate and noise level may depend on position. This allows for model features such as unstable (*Wong and Wang, 2006*; *Roxin and Ledberg, 2008*; *Erlich et al., 2015*) and leaky (*Wong and Wang, 2006*; *Ossmy et al., 2013*; *Farashahi et al., 2018*) integration. Leaky integration with a short time constant is functionally similar to the urgency-gated model with a low pass filter (*Hawkins et al., 2015a*; *Thura and Cisek, 2014*). Likewise, position-dependent variance may allow for multiplicative noise (*Freyer et al., 2012*). Third, the bound may change over time, which allows collapsing bounds according to an arbitrary function (*Hawkins et al., 2015a*; *Evans and Hawkins, 2019*; *Drugowitsch et al., 2012*), such as for an urgency signal. Fourth, models are parameterizable by an arbitrary dependence on fittable parameters or task conditions. These parameters may be shared between two related experimental tasks, such as an experimental and control task. It also allows alternative implementations of side bias (*Hanks et al., 2011*) and condition-dependent nonlinearities (*Shinn et al., 2020*). Fifth, across-trial parameter variability is supported within the GDDM framework for both starting point and non-decision time, but not drift rate.

Mathematically, these concepts are expressed using five functions, corresponding to the drift rate, noise, bound height, starting position, and post-factum modifications to the distribution (*Equation 1*). Collectively, the GDDM framework allows complicated models to be expressed in a consistent manner, simplifying the way which we may think about complicated DDM extensions.

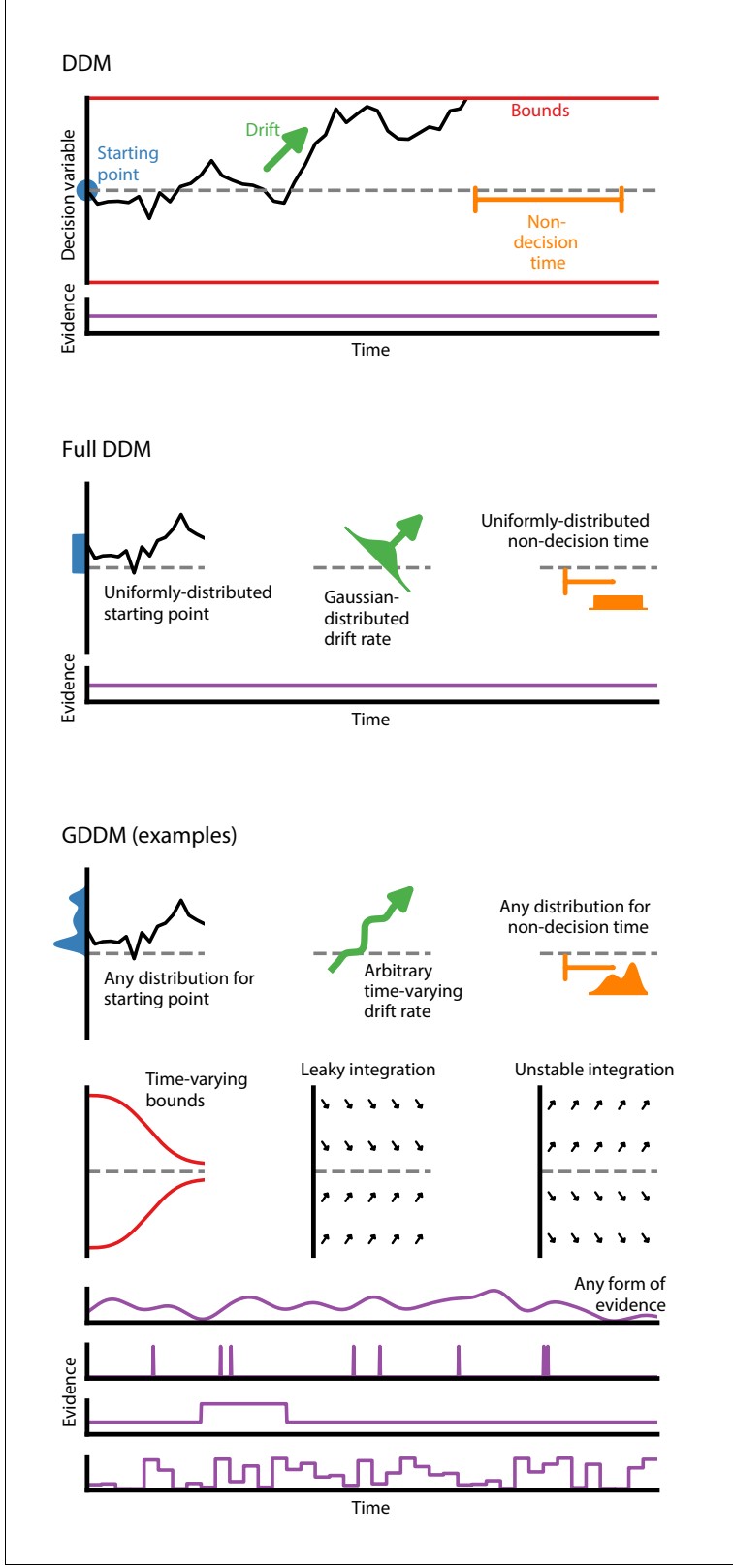

**Figure 1.** Generalized DDM. The DDM has fixed, limited parameters, and requires a stream of constant evidence. The 'full DDM' expands on the DDM by allowing uniformly-distributed starting position or non-decision time and Gaussian-distributed drift rate. The GDDM is a generalization of the DDM framework which allows arbitrary distributions for starting position and non-decision time, as well as arbitrary functions (instead of distributions) for

*Figure 1 continued on next page*

*Figure 1 continued*

drift rate and collapsing bounds. It also permits experimental paradigms with dynamic evidence. Several examples of potential GDDM mechanisms are shown. In each case, total response time is the time to reach the diffusion bound plus the non-decision time.

## GDDM mechanisms characterize empirical data

The cognitive mechanisms by which evidence is used to reach a decision is still an open question. One way to distinguish these mechanisms from one another is by fitting models of these cognitive mechanisms to empirical RT distributions. To demonstrate the flexibility of GDDMs, we constructed a GDDM including three plausible mechanisms which are difficult or impossible to implement in a standard DDM: collapsing bounds, leaky integration, and a parameterized nonlinear dependence of drift rate on stimulus coherence. As described below in more detail, we then tested the fit of a GDDM which includes these three mechanisms, compared to the DDM and the 'full DDM', in two psychophysical datasets from perceptual decision-making tasks: one from monkeys (*Roitman and Shadlen, 2002*), and one from human subjects (*Evans and Hawkins, 2019*). We found that these mechanisms allow the GDDM to provide a better fit to RTs with a more parsimonious model than the DDM or the 'full DDM' in the monkey dataset, suggesting that these mechanisms may be important to explaining the monkeys' behavior. However, consistent with previous results (*Hawkins et al., 2015a*), the GDDM fit no better than the DDM or 'full DDM' in the human dataset, suggesting that different cognitive strategies may be used in each case.

*Roitman and Shadlen, 2002* trained two rhesus monkeys on a random dot motion discrimination task in which dots moved randomly but with varying strengths of coherent motion to the left or right (*Figure 2—figure supplement 1*). Motion coherence was randomly selected to be one of six conditions ranging from 0% to 51.2%. The monkeys were required to determine the direction of coherent motion and respond via saccade to flanking targets. Response time was recorded, which we fit with the standard DDM, 'full DDM', and GDDM.

We compared the GDDM to the 'full DDM' and to three different implementations of the DDM which differ in the methodology used to estimate the RT distribution. A detailed comparison of these three packages—PyDDM, HDDM, and EZ-Diffusion—as well other common packages is provided in the section 'Software package for GDDM fitting'. The DDMs included terms for drift, non-decision time, and either bound height or noise magnitude. The DDM fit by PyDDM assumed drift rate was a linear combination of some baseline drift rate and coherence (*Equation 14*), whereas the DDM fit by HDDM used separate drift rates for each coherence condition (*Equation 16*), and EZ-Diffusion fit all parameters fully independently for each coherence condition (*Equation 17*). We also fit a 'full DDM' model, which included across-trial variability in the drift rate, non-decision time, and starting position (*Equation 15*). Finally, we fit a GDDM, which consisted of a typical DDM with no across-trial variability in the parameters, but which demonstrates three mechanisms that are difficult or impossible to model in a standard DDM framework: leaky integration, an exponentially collapsing bound, and a drift rate which is determined from coherence using a fittable nonlinearity (*Equation 13*). Each of the two monkeys was fit using a separate set of parameters for all models.

Each model was fit to the data, using full-distribution maximum likelihood for PyDDM and HDDM, and an analytic expression for EZ-Diffusion (*Figure 2*, *Figure 2—figure supplement 2*). First we examine the fit of the model quantitatively. Bayesian information criterion (BIC) is a common way of quantifying model fit while penalizing for free parameters. According to BIC, the 6-parameter GDDM fit better than all other models (*Figure 2a*). To determine whether this improved fit was due to the penalty BIC places on the number of parameters, we compared the models directly with likelihood. Again, we found that the GDDM fit this dataset better than the other models which have more parameters (*Figure 2b*). Finally, we tested whether this improved fit is due to the fact that PyDDM and HDDM use full-distribution maximum likelihood as an objective function, whereas EZ-Diffusion does not. We found that the mean squared error (MSE) of the GDDM was also lower than all other models (*Figure 2c*). These findings of a good fit by this GDDM, compared to the DDM and the 'full DDM', can be interpreted as evidence supporting the cognitive mechanisms included in the GDDM.

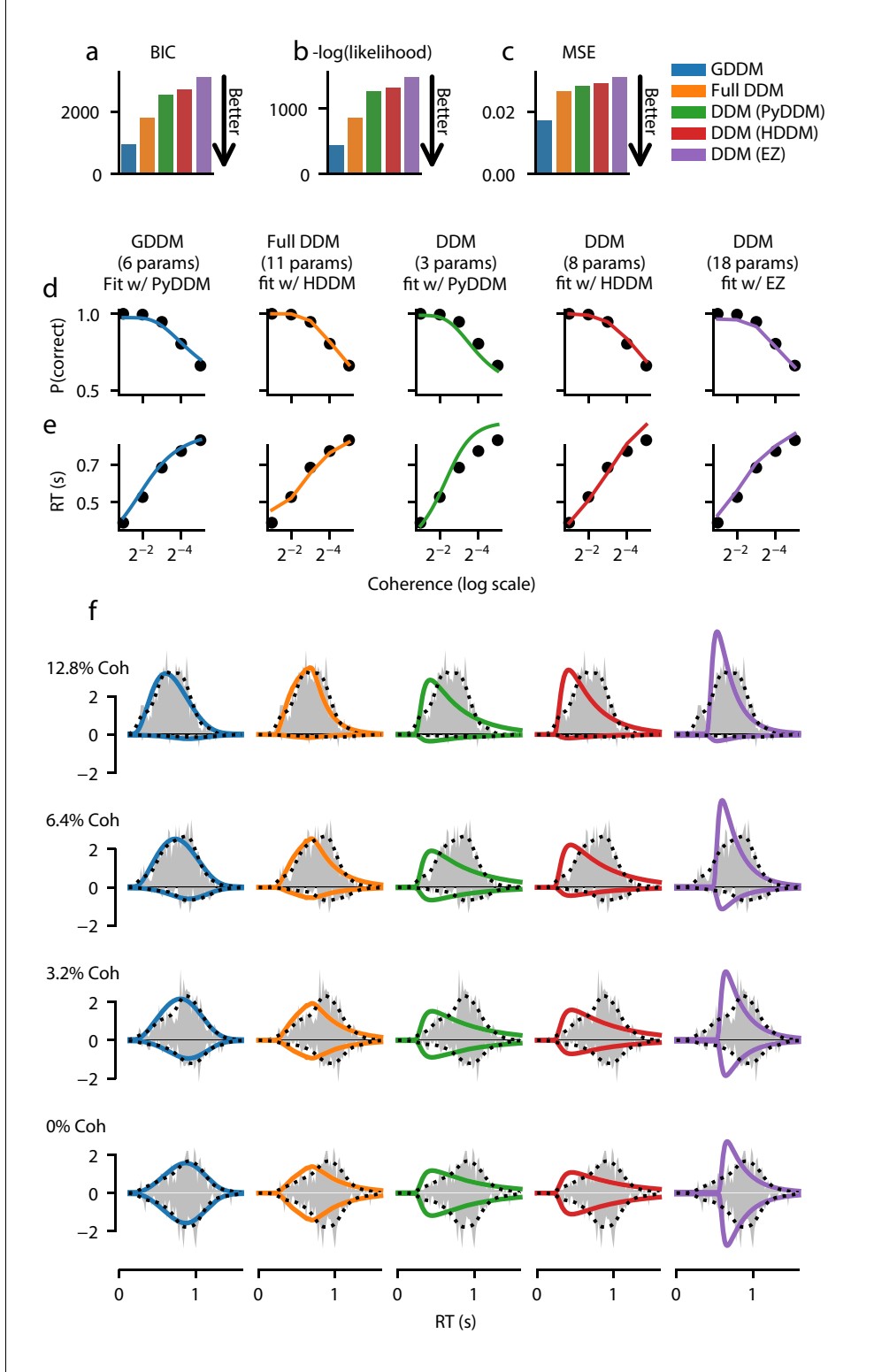

**Figure 2.** Fit of DDM, 'full DDM', and GDDM to *Roitman and Shadlen, 2002*. Five models (see **Materials and methods**) were fit with three different software packages to data from monkey N. Model fits are compared according to BIC (**a**), negative log likelihood (**b**), and mean squared error (MSE) (**c**). The psychometric (**d**) and chronometric (**e**) functions represent data as dots and models as lines. The probability density function (PDF) (**f**) of each model is shown for 12.8% coherence, 6.4% coherence, 3.2% coherence, and 0% coherence. The data RT

*Figure 2 continued on next page*

*Figure 2 continued*

histogram is shown in gray, and black dots show the smoothed kernel density estimate probability distribution derived from the data (*Scott, 2015*). Columns of the figure and the color of the line denote different models. The online version of this article includes the following figure supplement(s) for figure 2:

**Figure supplement 1.** Random dot motion task.
**Figure supplement 2.** Fit of DDM and GDDM in monkey B.
**Figure supplement 3.** Fit of DDM and GDDM to *Evans and Hawkins, 2019*.

Additionally, we assessed the qualitative fit of the models to the data to gain insight into what aspects of the RT distributions the GDDM was better able to capture. All five models provided a good fit to empirical psychometric (*Figure 2d* and *Figure 2—figure supplement 2d*) and chronometric (*Figure 2e* and *Figure 2—figure supplement 2e*) functions. However, models varied substantially in their ability to fit the qualitative properties of the RT distribution. The standard DDMs fit with HDDM and PyDDM showed a faster increase in response rate for the RT distribution at all coherence levels, and hence underestimated the mode of the distribution (*Figure 2f* and *Figure 2—figure supplement 2f*). The DDM fit with EZ-Diffusion overestimated the rate of increase in responses but did not begin to increase until the monkeys had made many responses, better matching the mode but creating an incorrect shape (*Figure 2f* and *Figure 2—figure supplement 2f*). The 'full DDM' fit better than the DDMs, and managed to capture the general shape of the RT distribution at high coherences (*Figure 2f* and *Figure 2—figure supplement 2f*), in addition to low coherences for one monkey (*Figure 2—figure supplement 2*). However, the GDDM fit better and with fewer parameters than the 'full DDM', matching the rate of increased responses and the overall distribution shape for all models under all conditions (*Figure 2f*, *Figure 2—figure supplement 2f*).

The RT distributions produced by monkeys may not be representative of those produced by humans (e.g., *Hawkins et al., 2015a*), which show a characteristic skewed distribution. This skew is characteristic of the DDM without collapsing bounds, and highlights that a different strategy may be used, potentially driven by overtraining or task structure (*Hawkins et al., 2015a*). We therefore fit the same GDDM to a more representative dataset from humans performing a similar perceptual decision-making task. Specifically, from the publicly available dataset of *Evans and Hawkins, 2019*, which used a random dot motion discrimination task (similar to *Roitman and Shadlen, 2002*), we fit the same set of models to the trials with no feedback delay, which the authors had found were better fit with no or little collapse of decision bounds. Consistent with the findings of *Hawkins et al., 2015a* and *Evans and Hawkins, 2019*, here we found that the 6-parameter GDDM provides no better fit than the 11-parameter 'full DDM' and 8-parameter DDM fit using HDDM (*Figure 2—figure supplement 3a–c*). The GDDM, 'full DDM', and DDMs fit with PyDDM and HDDM are all able to provide good fits to the psychometric function, chronometric function, and RT distributions for human subjects (*Figure 2—figure supplement 3d–f*). This lack of improved fit by the GDDM suggests that the human subjects employed a cognitive strategy closer to the standard DDM, and that the specific mechanisms of this GDDM (collapsing bounds, leaky integration, and input nonlinearity) are not important for explaining psychophysical behavior in this dataset.

For the example datasets considered above, the GDDM presented here improved the fit to the monkey data and provided a good fit to the human data. We caution that this result does not necessarily mean that this particular GDDM is appropriate for any particular future study. Models should not only provide a good fit to data, but also be parsimonious in form and represent a clear mechanistic interpretation (*Vandekerckhove et al., 2015*). The examples here demonstrate the utility of GDDMs for quantitatively evaluating cognitive mechanisms of interest through fitting models to empirical data.

## Methods for estimating RT distributions in the DDM and GDDM

There are three broad classes of methods for estimating a GDDM's RT distribution. First, analytical expressions can be derived using stochastic calculus and sequential analysis (*Wald, 1945*; *Anderson, 1960*). Analytical expressions were favored historically because they can be computed quickly by humans or computers. However, apart from the simplest form of the DDM and a few particular classes of extensions, analytical solutions often do not exist. While analytical expressions offer

maximum performance and accuracy, deriving new analytical expressions is not a practical way to rapidly implement and test new GDDMs.

The second method, trial-wise trajectory simulation, fully supports GDDM simulation. This method simulates the decision variable trajectories of many individual trials to sample RTs without producing a probability distribution (*Figure 3a*). Previous work has used trial-wise trajectories to simulate DDMs with no known analytical solution (e.g. *Hawkins et al., 2015a*). However, this flexibility comes at the cost of performance. As we show in section 'Execution time vs error tradeoff', trial-wise trajectory simulation is very slow and provides a coarse-grained representation of the model. Since this method does not produce a probability distribution, it is difficult to use efficient and robust methods such as full-distribution maximum likelihood for fitting data, though progress is ongoing in this area using likelihood-free methods such as through approximate Bayesian computation (*Holmes and Trueblood, 2018*).

By contrast, the third method may use the Fokker-Planck equation to numerically compute the first passage time distribution. The Fokker-Planck equation is a mathematical expression which reformulates the trial-by-trial diffusion process: rather than simulating single-trial decision variable trajectories, this method 'simulates' all possible paths at once by propagating the distribution of single-trial decision variable trajectories over time (*Figure 3b*). Unlike analytical solutions, numerical solutions from the Fokker-Planck equation can be computed for all GDDMs. They also generate a probability density function for the RT distribution instead of a sample, allowing fitting with full-distribution likelihood methods. Several algorithms may be used to solve the DDM or GDDM using the Fokker-Planck equation, each with different advantages and disadvantages.

We consider three such algorithms: forward Euler, backward Euler, and Crank-Nicolson (see Appendix). These algorithms differ in how they propagate the probability distribution over time: forward Euler iteratively approximates the probability distribution of trajectory position at each timestep using the distribution at the previous timestep, while backward Euler and Crank-Nicolson iteratively solve systems of linear equations to approximate the probability distribution (*Voss and Voss, 2008*). The three algorithms also differ in their conceptual and technical difficulty to implement, with forward Euler being easier than backward Euler or Crank-Nicolson. In section 'Execution time vs error tradeoff', we will show that the backward Euler and Crank-Nicolson algorithms provide higher accuracy for a given execution time—and a faster execution time for a given accuracy level—than the forward Euler algorithm and trial-wise trajectory simulation.

## Execution time vs error tradeoff

Next, we evaluated the effectiveness of different methodologies for estimating the RT distribution. Two general methodological considerations for all of these models are the precision with which the simulations are performed (accuracy), and the amount of time it takes the simulation to run on a computer (execution time). Accurate RT distributions are important to ensure the result faithfully represents the model, and fast execution times are important to increase the feasibility of fitting models to data. In general, for a given algorithm, there is a tradeoff between execution time and

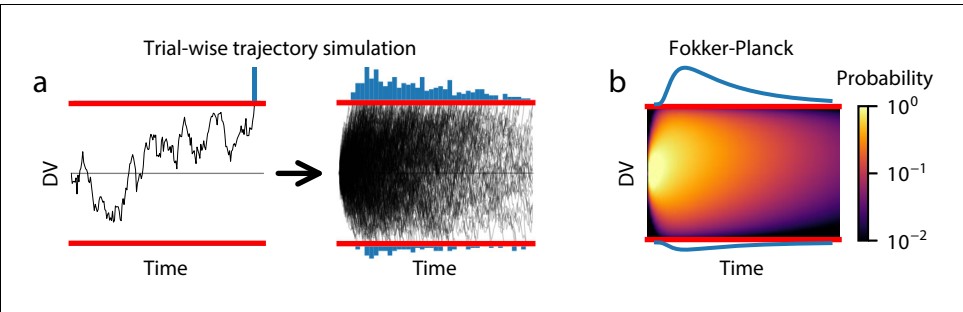

**Figure 3.** Methods for estimating GDDM RT distributions. (**a**) A schematic of trial-wise trajectory simulation methods. In a single trial, a subject's decision process can be summarized by a stochastic decision variable (DV); simulating this single-trial decision variable many times results in a histogram approximation to the RT distribution. (**b**) A schematic of solving the Fokker-Planck equation. The full probability distribution of decision variables is solved at each time point, providing a numerical solutions with higher accuracy and reduced execution time.

the accuracy of the solution obtained. One algorithm may be more efficient than another algorithm if it produces similar accuracy in a shorter amount of time, or better accuracy in the same amount of time.

We directly evaluate the execution time vs error tradeoff in trial-wise trajectory simulation and in solving the Fokker-Planck equation. To do so, we consider a standard DDM with constant drift, diffusion, and bounds (*Equation 18*). While this is not itself a GDDM, such a model has an analytical solution to the reaction-time distribution (as a closed-form infinite sum) (*Anderson, 1960*; *Ratcliff, 1978*) which can be used as ground truth for computing the error in numerical estimates (see **Materials and methods**).

First, we test trial-wise trajectory simulation. We perform a parameter sweep across timesteps $\Delta t$ and sample sizes $N$. We simulate using the Euler-Maruyama method, which is standard in the DDM literature (e.g. *Hawkins et al., 2015a*; *Hawkins et al., 2015b*; *Evans et al., 2017*; *Harty et al., 2017*; *Zhou et al., 2009*; *Atiya et al., 2020*; *Simen et al., 2011*; *Nguyen et al., 2019*; *Lewis et al., 2014*; *Verdonck et al., 2016*) and for simple models provides identical truncation error but better execution time than more sophisticated methods such as stochastic fourth-order Runge-Kutta (*Gard, 1988*; *Brown et al., 2006*). Because decreasing $\Delta t$ for a fixed $N$ increases the number of timebins and thus increases the variability across each timebin, we employed a smoothing procedure on the resulting simulated histogram to account for this. For each simulation, we smoothed the simulated RT histogram with all mean filters of size 1 to 500 points and chose the filter with the lowest error; since a filter of size one is equivalent to applying no smoothing, this procedure could only improve accuracy. In practice, selecting a filter this way would not be possible due to the lack of analytical ground truth for most GDDMs, and so this smoothing procedure artificially inflates the method's performance compared to other methods, and effectively yields an upper bound on trial-wise trajectory performance. Execution time and error are shown separately for a range of numerical parameters (*Figure 4a,b*). As a proxy for the execution time vs error tradeoff, we also show the product of execution time and mean squared error for the parameter sweep (*Figure 4—figure supplement 1*).

In general, trial-wise trajectory simulation provides a suboptimal execution time vs error tradeoff. Using the product of MSE and execution time as a proxy measure of overall performance to illustrate this tradeoff, trial-wise trajectories perform most efficiently for highly imprecise models, i.e. those with a large timestep and a small number of trajectories simulated (*Figure 4—figure supplement 1*). To examine the execution times and accuracies which lead to this tradeoff, we fix $\Delta t = 0.01$ and plot the execution time and accuracy across many different values of sample size $N$ (*Figure 4c*). We see that as execution time increases, accuracy does not increase by more than one order of magnitude. Examples of these traces are provided in *Figure 4—figure supplement 2*.

Next, we examine methods for solving the Fokker-Planck equation. We consider three different algorithms: forward Euler, backward Euler, and Crank-Nicolson (*Table 1*). The execution time vs accuracy tradeoff of these three algorithms are governed by two parameters: the timestep $\Delta t$, and the granularity of decision variable discretization $\Delta x$. Decreasing either of these two parameters slows down the execution time but increases accuracy. These three algorithms operate in a distinct but related manner. Forward Euler is the easiest algorithm to implement, however it is only numerically stable for values of $\Delta t$ which are very small relative to $\Delta x$. Forward Euler is an approximation in the limit as $\Delta t \to 0$, and thus cannot be run for large $\Delta t$ step sizes relative to $\Delta x$ (see **Materials and methods**). Backward Euler solves this problem by providing a numerically stable solution across all values of $\Delta t$ and $\Delta x$. While it provides a similar execution time and accuracy as forward Euler given $\Delta t$ and $\Delta x$, the fact that it is numerically stable across all values of these parameters allows more efficient values of $\Delta t$ and $\Delta x$ to be chosen. Crank-Nicolson further improves the accuracy and execution time over backward Euler (*Table 1*). Like backward Euler, Crank-Nicolson does not have theoretical constraints on $\Delta x$ or $\Delta t$, but unlike backward Euler, it may experience numerical instability for models with time-variant integration bounds. In practice, for each of these methods, the required maximum step size is constrained by how strongly varying the drift and noise are in space and time, where more varying drift would require a smaller step size. Convergence of the results can be evaluated by checking results with smaller step sizes.

To evaluate the execution time vs error tradeoff, we perform a parameter sweep across values of $\Delta t$ and $\Delta x$ for these three algorithms and again compute the product of execution time and mean squared error as a proxy for execution time vs error tradeoff (*Figure 4a,b*, *Figure 4—figure*

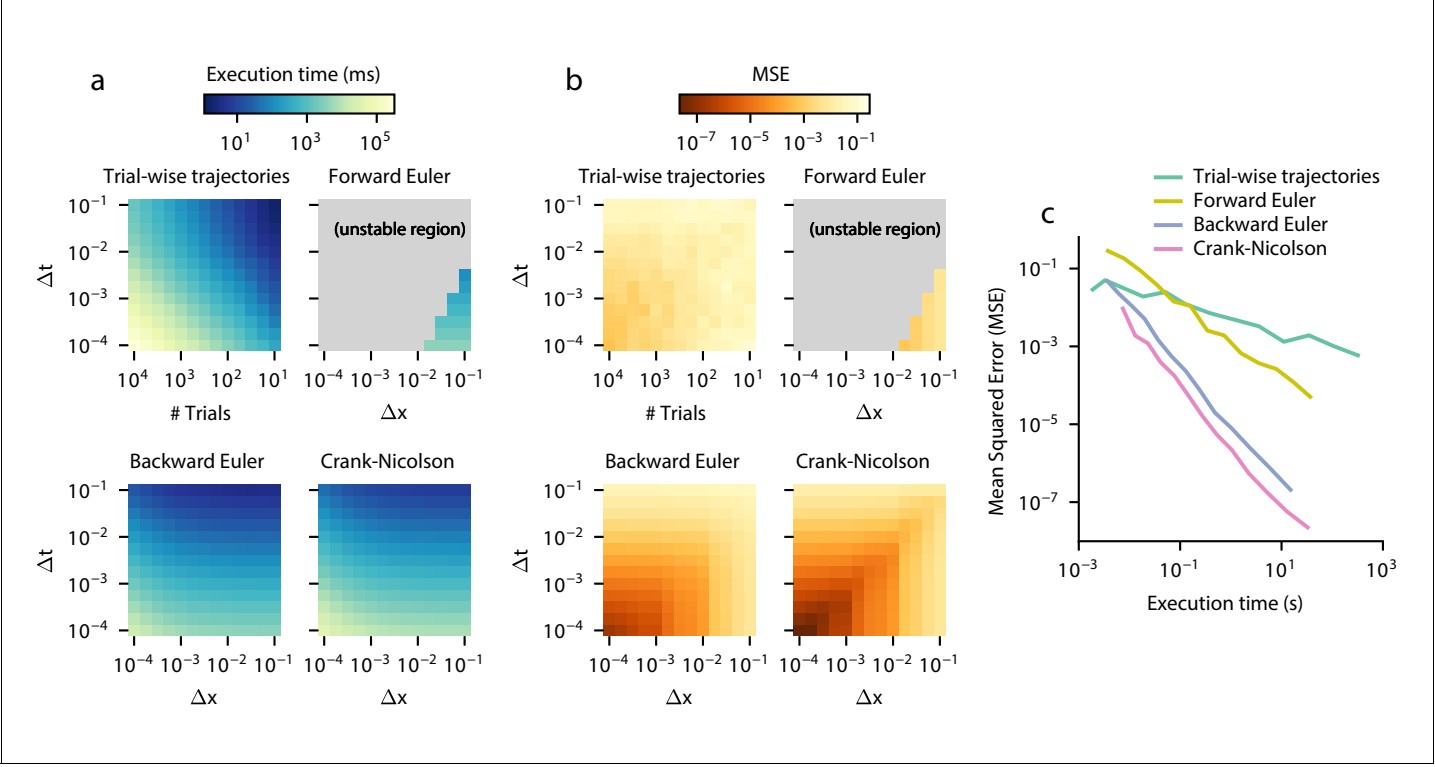

**Figure 4.** Execution time vs error tradeoff. A DDM with constant drift, noise, and bounds was estimated for different spatial ($\Delta x$) and temporal ($\Delta t$) resolutions. Execution time (**a**) and MSE (**b**) are shown across values of $\Delta x$ and $\Delta t$. MSE was computed against the analytical DDM solution for the full distribution. Darker colors indicate a more favorable execution time vs error tradeoff. Gray color indicates numerical instability. (**c**) Relationships between execution time and error which approximately maximize this tradeoff are shown. Parametric expressions were empirically determined to approximately maximize accuracy vs error tradeoff (see *Figure 4—figure supplement 1*), and are given by $\Delta t = 0.43\Delta x^2$ for forward Euler and $\Delta t = \Delta x$ for trial-wise trajectory simulation, backward Euler, and Crank-Nicolson. The lower left indicates low error and fast execution time whereas the upper right indicates slow execution time and high error.

The online version of this article includes the following figure supplement(s) for figure 4:

**Figure supplement 1.** Execution time vs MSE tradeoff across parameters.

**Figure supplement 2.** Example estimated RT distributions with different numerical parameters.

*supplement 1*). Overall, the execution time vs error tradeoff is one to six orders of magnitude better for these algorithms than for trial-wise trajectory simulation. We find that, for backward Euler and Crank-Nicolson, the execution time vs error tradeoff in the model shown in *Figure 4—figure supplement 1* is maximized approximately when $\Delta t = \Delta x$. Forward Euler is unstable in this parameterization, with the best execution time vs error tradeoff when it is closest to its stability condition. We

**Table 1.** Methods for estimating RT distributions, in order of priority within PyDDM.

| | Time accuracy | Grid accuracy | Requirements |
|---|---|---|---|
| Analytical | Exact | Exact | Time- and position-independent drift rate, diffusion constant, and bounds (or linearly collapsing bounds); centered point source initial condition |
| Crank-Nicolson | $O(\Delta t^2)$ | $O(\Delta x^2)$ | Time independent bounds |
| Backward Euler | $O(\Delta t)$ | $O(\Delta x^2)$ | None (fallback for all models) |
| | | | |
| Forward Euler | $O(\Delta t)$ | $O(\Delta x^2)$ | Numerics $\Delta t/\Delta x^2 < 1/\sigma^2$ (not used by PyDDM) |
| Trial-wise trajectories | N/A | N/A | (not used by PyDDM) |

examine the execution time and error for parameterizations which satisfy these constraints, and compare them to trial-wise trajectory simulation (*Figure 4c*). We see that Crank-Nicolson is strictly better than other methods: for a given error, Crank-Nicolson offers a faster execution time, and for a given execution time, Crank-Nicolson offers lower error. However, since Crank-Nicolson may not be appropriate for models with time-variant boundaries, the second best method is also of interest. We see that, for a given execution time, backward Euler is better than forward Euler by two orders of magnitude, and better than trial-wise trajectory simulations by up to five orders of magnitude. Therefore, solving the Fokker-Planck equation using Crank-Nicolson and backward Euler represents a key advantage over the forward Euler and trial-wise trajectory simulation methods.

## Software package for GDDM fitting

### Overview of our software package

We developed a software package to encapsulate the methodological framework we have presented. Our package, PyDDM, allows the user to build models, simulate models, and fit them to data using the efficient GDDM framework methodology described previously. The structure and workflow of the package were designed to mirror the GDDM framework (*Figure 5—figure supplement 1*). PyDDM selects the most efficient approach to solve a GDDM by automatically determining whether the model can be solved analytically and whether it uses time-varying bounds (*Table 1*). PyDDM is also built to emphasize the modularity of the GDDM framework. Components of models may be readily reused in other models, and to promote this, we additionally provide an online database of models (the 'PyDDM Cookbook' https://pyddm.readthedocs.io/en/latest/cookbook/index. html) to promote code sharing and replication. Users are encouraged to submit their models to this database.

Furthermore, PyDDM is designed to accommodate the arbitrary flexibility of GDDMs. For reliable parameters estimation, it supports differential evolution (*Storn and Price, 1997*) as a global optimization method in addition to local search methods such as the Nelder-Mead simplex algorithm (*Nelder and Mead, 1965*). Fits are performed using full-distribution maximum likelihood on the full probability distribution in order to use all available data to estimate parameters. Likelihood can be overly sensitive to outliers or 'contaminants' (*Ratcliff and Tuerlinckx, 2002*), and this problem has been solved elsewhere by using a mixture model of the DDM process with a uniform distribution in a fixed ratio (*Wiecki et al., 2013*; *Wagenmakers et al., 2008b*; *Voss and Voss, 2007*). PyDDM builds on this methodology by allowing a parameterizable mixture model of any distribution and a fittable ratio. Because some models can be difficult to understand conceptually, PyDDM also includes a GUI for visualizing the impact of different model parameters (*Figure 5—figure supplement 2*), analogous to a GDDM-compatible version of the tool by *Alexandrowicz, 2020*.

### Comparison to other software packages

Besides PyDDM, several software packages are available to simulate DDMs, and one package to simulate GDDMs. We compare these in detail in *Table 2*, to emphasize a few key points of comparison.

The choice of software package depends on the model to be fit. As demonstrated in *Figure 2* and noted previously (*Ratcliff, 2008*), fitting a diffusion model to a small number of summary statistics, as in EZ-Diffusion, can lead to poor qualitative and quantitative model fit to the RT distribution. Better options for fitting the DDM and 'full DDM' are available, such as fast-dm and HDDM. However, these packages are limited in model form, and make it difficult to extend beyond the 'full DDM' to test additional cognitive mechanisms or harness time-varying stimulus paradigms. As we have shown previously (*Figure 2*), the GDDM is desirable for obtaining good model fits with few parameters.

The major highlight of PyDDM compared to other packages is that it allows GDDMs to be simulated, and is thus compatible with time- and position-dependent drift rate and noise, time-varying bounds, starting point variability, and non-decision time variability. Besides PyDDM, the only other package to our knowledge capable of simulating GDDMs is the new R package CHaRTr (*Chandrasekaran and Hawkins, 2019*). This package is a major innovation compared to previous approaches, as it allows flexible model forms to be tested. However, CHaRTr is based on trial-wise trajectory simulations. As shown previously (*Figure 4*), trial-wise trajectory simulation is especially

**Table 2.** Comparison of existing DDM packages.

PyDDM is compared to HDDM (*Wiecki et al., 2013*), EZ-Diffusion (*Wagenmakers et al., 2007*), CHaRTr (*Chandrasekaran and Hawkins, 2019*), Diffusion Model Analysis Toolbox (DMAT) (*Grasman et al., 2009*; *Vandekerckhove and Tuerlinckx, 2008*), and fast-dm (*Voss and Voss, 2007*; *Voss and Voss, 2008*; *Voss et al., 2015*). Red indicates limited flexibility, yellow indicates moderate flexibility, and green indicates maximal flexibility. For the solver, these colors indicate minimal, moderate, and maximal efficiency.

| | PyDDM | HDDM | EZ-Diffusion | CHaRTr | DMAT | fast-dm |
|---|---|---|---|---|---|---|
| Language | Python3 | Python2/3 | Matlab, R, Javascript, or Excel | Requires both R and C | Matlab | Command line |
| Solver | Fokker-Planck, analytical | Analytical numerical hybrid | None | None (Monte Carlo) | Analytical numerical hybrid | Fokker-Planck |
| **Task parameters** | | | | | | |
| Time dependence of drift/noise | Any function | Constant | Constant | Any function | Constant | Constant |
| Position dependence of drift/noise | Any function | Constant | Constant | Any function | Constant | Constant |
| Bounds | Any function | Constant | Constant | Any function | Constant | Constant |
| Parameter dependence on task conditions | Any relationship for any parameter | Regression model | Categorical | Categorical | Linear | Categorical |
| **Across-trial variability** | | | | | | |
| Across-trial drift variability | Slow discretization (via extension) | Normal distribution | None | Any distribution | Normal distribution | Normal distribution |
| Across-trial starting point variability | Any distribution | Uniform distribution | None | Any distribution | Uniform distribution | Uniform distribution |
| Across-trial non-decision variability | Any distribution | Uniform distribution | None | Any distribution | Uniform distribution | Uniform distribution |
| **Model simulation and fitting** | | | | | | |
| Hierarchical fitting | No | Yes | No | No | No | No |
| Fitting methods | Any numerical (default: differential evolution) | MCMC | Analytical | Any numerical | Nelder-Mead | Nelder-Mead |
| Objective function | Any function (default: likelihood) | Likelihood | Mean/stdev RT and P(correct) | Any sampled (e.g. quantile maximum likelihood) | Quantile maximum likelihood or chi-squared | Likelihood, chi-squared, Kolmogorov-Smirnov |
| Mixture model | Any distribution(s) | Uniform | None (extendable) | None | Uniform and undecided guesses | Uniform |

inefficient, reducing the accuracy of the model simulations and increasing execution time by several orders of magnitude. In practice, this makes it infeasible to fit data to more flexible models. For the user to define new models, CHaRTr requires defining models in C and modifying the CHaRTr source code. By contrast, PyDDM facilitates the specification and implementation of models by defining them in user scripts written in pure Python.

## Parallel performance

In addition to PyDDM's efficient algorithms for solving GDDM models, PyDDM also includes the ability to further speed up simulations by solving models in parallel. Any model can be converted from single-threaded to multi-threaded with a single function call. PyDDM solves models for different conditions on multiple CPUs via the 'pathos' library (*McKerns et al., 2012*). It uses a so-called 'embarrassingly parallel' algorithm, which turns the most essential parts of the simulation into independent tasks and distributes them equally across CPUs. Such parallelization can be enabled in any PyDDM script with one line of code.

In a perfectly efficient parallelized environment, a simulation utilizing $N$ CPUs should be $N$ times faster than a simulation utilizing one CPU. In practice, this is rare due to communication latency and

other overhead. We evaluate the speedup gained by solving an example DDM (*Equation 19*)) on more than one CPU (*Figure 5*). As expected, the execution time decreases when more CPUs are used to perform the simulations. However, this speedup is strikingly linear (*Figure 5a*); for up to 20 CPUs, PyDDM achieves close to 100% parallel efficiency (*Figure 5b*). This means that PyDDM is able to effectively utilize multiple processors to decrease execution time. In practice, the 6-parameter GDDM shown above fits on one CPU in under 15 minutes, and the authors routinely fit 15-parameter GDDMs on 8 CPUs with differential evolution for global parameter optimization in under 5 hours.

## Discussion

The GDDM framework provides a consistent description of model extensions, and allows researchers to efficiently simulate and fit a wide variety of extensions to the DDM. Using two open datasets, we demonstrated that GDDMs are able to fit experimental data with high accuracy and few parameters. Model RT distributions are solved and fit numerically using multiple fast and accurate methods for solving the Fokker-Planck equation. We built the PyDDM software package to implement the GDDM framework. PyDDM makes it easy to utilize the efficiency of this framework, while providing additional technical advantages over other packages.

The major benefit of our GDDM framework is the flexibility and ease with which the model may be extended. The GDDM generalizes the DDM to support a wide variety of task paradigms and cognitive mechanisms. Any parameter in the standard DDM is allowed to be an arbitrary function of time and/or position when applicable. Mechanisms such as leaky or unstable integration, collapsing bounds, time-dependent drift rates, and complex non-decision time and starting point distributions all fall within the GDDM framework and can be readily simulated in PyDDM. GDDMs may be built modularly, eliminating the need to reimplement a separate model for each new mechanism. This modularity also helps promote code reuse through our online database of PyDDM model components.

GDDMs may be simulated by efficiently solving the Fokker-Planck equation. In our tests, trial-wise trajectory simulation performed poorly because large increases in the number of trials did not result in as large of increases in accuracy. While the smoothing procedure we applied improved accuracy, trial-wise trajectory simulation was unable to obtain low error for reasonable execution times. The Crank-Nicolson and backward Euler algorithms performed up to five orders of magnitude better than trial-wise trajectory simulation. This increase in performance is consistent with previous work which used Crank-Nicolson to solve the Fokker-Planck equation for the 'full DDM' (*Voss and Voss, 2008*). However, increasing performance is critical for the GDDM not only because it reduces

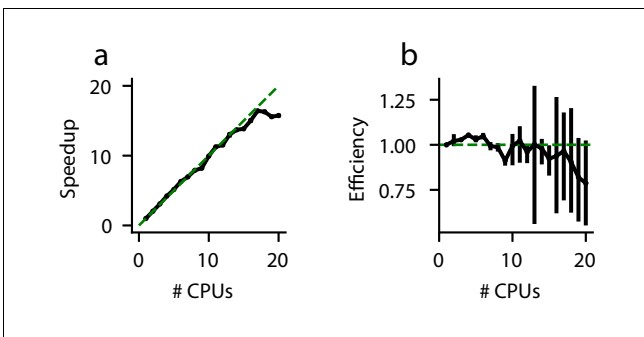

**Figure 5.** Parallel performance. Simulations of a standard DDM were performed, varying numbers of CPUs each with ten replicates. (a) The speedup, defined as (execution time on 1 CPU)/(execution time on *N* CPUs), and (b) parallel efficiency, defined as speedup/(*N* CPUs), are shown for different numbers of CPUs. Because measured execution time varied run to run, the mean parallel efficiency could sometimes exceed 1. Error bars are bootstrapped 95% confidence intervals. Solid black lines indicate data, and dashed green lines indicate the theoretical maximum under noiseless conditions. Confidence intervals in (a) are hidden beneath the markers. The online version of this article includes the following figure supplement(s) for figure 5:

**Figure supplement 1.** Organization of PyDDM's source code.
**Figure supplement 2.** Model GUI.

simulation time, but because it leads to a qualitative difference in the types of models which may be fit to data. While inefficient methods are sufficient for simulating a single instance of the model, fitting models with many parameters to data requires thousands of simulations or more. By providing efficient algorithms for solving this broad class of models, our framework makes it possible to fit parameters to data for models which were previously prohibitively time consuming. In this way, it expands the range of potential cognitive mechanisms which can be tested and experimental paradigms which can be modeled.

PyDDM enables researchers to build very complex models of decision-making processes, but complex models are not always desirable. Highly complex models may be able to provide an excellent fit to the data, but unless they represent specific, plausible cognitive mechanisms, they may not be appropriate. PyDDM's complexity can be used to examine a range of potential cognitive mechanisms which were previously difficult to test, and to probe a range of task paradigms. Additionally, sometimes complex models which do represent appropriate cognitive mechanisms may exhibit inferior performance due to limitations of the available data. For instance, simulation studies suggest that when data are sparse, Bayesian fitting or EZ-Diffusion may be better able to recover parameters (*Lerche et al., 2017*; *Wiecki et al., 2013*; *van Ravenzwaaij et al., 2017*).

When the hypothesis demands additional model complexity, though, one critical check is to make sure the parameters of the models are recoverable. That is, given a set of models, simulated data from each model should be best fit by the model and parameters that simulated it (*Wiecki et al., 2013*). This explicitly tests against model and parameter degeneracy, a known weakness of the 'full DDM' (*van Ravenzwaaij and Oberauer, 2009*; *Boehm et al., 2018*). Individual GDDMs must be tested for model and parameter recovery before claims can be made about model mechanisms. PyDDM makes it straightforward to test whether a given GDDM has parameters which are recoverable, and whether any two given models can be distinguished from one another. This is facilitated because synthetic RT data can be sampled from a the calculated probability distributions of the GDDM without the need to run trial-wise trajectory simulations.

The GDDM framework supports across-trial variability in non-decision time and starting position according to any arbitrary distribution, but it does not support across-trial drift rate variability. Across-trial drift rate variability is an inherent weakness of the Fokker-Planck approach. It is technically possible to model across-trial drift rate variability using our framework, by discretizing the drift rate distribution and solving the Fokker-Planck equation for each case separately (*Voss and Voss, 2008*). PyDDM is not set up to ergonomically reconcile across-trial drift rate variability with time-varying evidence, which is a core feature of PyDDM. As a result, the GDDM framework does not fully support the across-trial drift rate variability aspect of the 'full DDM'. Across-trial drift rate variability is a potentially plausible cognitive mechanism used in prior literature to account for slow errors (*Ratcliff and Rouder, 1998*). Of note, studies suggest this mechanism may not be necessary to include to fit data (*Lerche and Voss, 2016*; *Lerche et al., 2017*). GDDMs offer researchers the capability to explore alternative cognitive mechanisms which might also account for slow errors.

PyDDM uses numerical solutions of the Fokker-Planck equation to estimate the RT distribution of the model. Other methods may be used for this estimation, such as solving the Volterra integral equation (*Smith, 2000*). Some prior work has used these alternative methods for solving GDDMs (*Drugowitsch et al., 2012*; *Zhang et al., 2014*; *Drugowitsch et al., 2014c*). Such methods already have fast solvers (*Drugowitsch, 2016*), which could be implemented as an alternative method within PyDDM in the future.

PyDDM also has the potential to support other applications of interest in the field. First, it could be used as a simulation engine for fitting hierarchical models of GDDM parameters, in which each subject's individual parameters are drawn from a group distribution. The most critical component to hierarchical model fitting is an efficient method for simulating model likelihood. Since PyDDM provides such a method, hierarchical fitting of GDDMs could be readily implemented as an extension to PyDDM. Second, while PyDDM currently only implements bounds which are symmetric around zero, support for asymmetric bounds is a feasible future addition. Third, PyDDM may be extended to support multi-dimensional diffusion, enabling it to represent interacting race models (*Purcell et al., 2010*; *Usher and McClelland, 2001*; *Bogacz et al., 2006*; *Tsetsos et al., 2012*; *Drugowitsch et al., 2014b*). The Fokker-Planck equation can be solved in higher dimensions, in principle enabling similar highly efficient methods to be used for these models as well. The GDDM framework would require only minor adaptations to be compatible with multi-dimensional diffusion. Fourth, it may be

extended to allow fitting models to neural recordings or other data which represents the value of the decision variable, as in *Zoltowski et al., 2020*. Currently, PyDDM only supports fitting to the RT distribution.

In summary, we show that the GDDM framework is a flexible and efficient way to explore new extensions on the DDM, allowing new possibilities in task design. Our package, PyDDM, offers a convenient and modular method for implementing this framework.

# Materials and methods

## GDDM description

The GDDM is described as the following differential equation for decision variable *x*:

$$dx = \mu(x, t, \ldots)dt + \sigma(x, t, \ldots)dW \tag{1}$$

where '...' represents task conditions and fittable parameters. Initial conditions are drawn from the distribution $x \sim X_0(\ldots)$. The process terminates when $|x(t, \ldots)| \geq B(t, \ldots)$. This can be parameterized by five functions:

- $\mu(x, t, \ldots)$ – The instantaneous drift rate, as a function of the decision variable position, time, and task parameters. In PyDDM, it is represented by the 'Drift' class and defaults to $\mu(x, t, \ldots) = 0$.
- $\sigma(x, t, \ldots)$ – The instantaneous noise, as a function of the decision variable position, time, and task parameters. In PyDDM, it is represented by the 'Noise' class and defaults to $\sigma(x, t, \ldots) = 1$.
- $X_0(\ldots)$ – The probability density function describing the initial position of the decision variable, as a function of task parameters. It must hold that $\sum_x X_0(x, \ldots) = 1$. In PyDDM, it is represented by the 'IC' class and defaults to a Kronecker delta function centered at $x = 0$.
- $B(t, \ldots)$ – The boundaries of integration, as a function of time and task parameters. In PyDDM, it is represented by the 'Bound' class and defaults to $B(t, \ldots) = 1$.
- $p_C^*(t, \ldots)$ and $p_E^*(t, \ldots)$ – Post-processing performed on the simulated first passage time distributions of correct and error RT distributions $p_C(t)$ and $p_E(t)$, respectively. In PyDDM, these functions are represented by the 'Overlay' class and default to $p_C^*(t, \ldots) = p_C(t)$ and $p_E^*(t, \ldots) = p_E(t)$. In practice, 'Overlay' is used to implement non-decision times and mixture models for fitting contaminant responses.

The organization of the code in PyDDM mirrors this structure (*Figure 5—figure supplement 1*).

The GDDM framework supports Markovian processes, in that the instantaneous drift rate and diffusion constant of a particle is determined by its position $x_t$ and the current time *t*, without dependence on the prior trajectory leading to that state. Drift rate is discretized, such that the timestep of the discretization is equal to the timestep of the Fokker-Planck solution. In practice, for typical task paradigms and dataset properties, the impact of this discretization is expected to be minimal for a reasonable timestep size.

## Fokker-Planck formalism of the GDDM

PyDDM solves the GDDM using the Fokker-Planck equation, a partial differential equation describing the probability density $\rho(x, t, \ldots)$ of the decision variable to be at position *x* at time *t*, with initial condition $X_0(\ldots)$:

$$\frac{\partial}{\partial t}\rho(x, t, \ldots) = -\frac{\partial}{\partial x}(\mu(x, t, \ldots)\rho(x, t, \ldots)) + \frac{\partial^2}{\partial x^2}\left(\frac{\sigma^2(x, t, \ldots)}{2}\rho(x, t, \ldots)\right). \tag{2}$$

By definition, the GDDM has absorbing boundary conditions ($\rho(x, t, \ldots) = 0$ for all $x \leq -B(t, \ldots)$ and $x \geq B(t, \ldots)$), and any mass of the probability density crossing the boundaries are treated as committed, irreversible decisions.

The Fokker-Planck equation permits numerical solutions of the GDDM. The numerical algorithm is described for the GDDM with fixed bounds in this section, and the algorithm for the GDDM with time-varying bounds is described in the next section. The system is first discretized with space and time steps $\Delta x$ and $\Delta t$. Without loss of generality, we assume $\Delta x$ divides $2B(t, \ldots)$ and $\Delta t$ divides the

simulated duration. The probability density solution of *Equation 2*, $\rho(x,t,\ldots)$, can then be approximated as the probability distribution function at spatial grids $\{-B(t,\ldots)+\Delta x, -B(t,\ldots)+2\Delta x, \ldots, 0, \ldots, B(t,\ldots)-2\Delta x, B(t,\ldots)-\Delta x\}$ and temporal grids $\{n\Delta t\}$ for any non-negative integer $n$. Due to absorbing bounds, the two spatial grids one step outward at $\pm B(t,\ldots)$ have 0 density, and are not considered explicitly. The choice of $\Delta t$ is constrained by the stability criteria and accuracy vs execution time tradeoff (see *Figure 4*).

Define $P_i^m$ as the probability distribution at the $i$th space-grid and the $m$th time-grid. For clarity, here and below $P_i^m$ denotes the total probability inside a grid of lengths $\Delta x$ and $\Delta t$, and its center represented by $(i,m)$. As such, $P_i^m$ is unitless, in contrast to $\rho(x,t)$ which has a unit of the inverse of the spatial dimension. Probability distribution functions can also be viewed as a discrete proxy of the continuous probability density (i.e. approximating $\rho(x,t)$ at the corresponding space and time grid), which will then have a unit of the inverse of the spatial dimension. Both views are mathematically equivalent (and just off by a $\Delta x$ factor to the whole equation), but the former definition was used for simplicity and interpretability.

The Fokker-Planck equations can be discretized in several distinct ways, including the forward Euler method, the backward Euler method, and the Crank-Nicolson method. In the forward Euler method, *Equation 2* is approximated as:

$$\frac{P_j^n - P_j^{n-1}}{\Delta t} = -\frac{(\mu P)_{j+1}^{n-1} - (\mu P)_{j-1}^{n-1}}{2\Delta x} + \frac{(DP)_{j+1}^{n-1} + (DP)_{j-1}^{n-1} - 2(DP)_j^{n-1}}{\Delta x^2} \tag{3}$$

where $D = \sigma^2/2$ is the diffusion coefficient. Here and below, dependencies of space, time, and other variables are omitted for simplicity. Terms in parenthesis with subscripts and superscripts are evaluated at the subscripted space grid and superscripted time grid, in the same manner as $P_j^n$. The spatial derivatives at the right side of *Equation 2* are evaluated at the previous time step $n-1$, such that only one term of the equation is at the current time step $n$, generated from the temporal derivative at the left side of *Equation 2*. As such, *Equation 3* for each $j$ fully determines $P_j^n$ from the probability distributions at the previous time-step:

$$P_j^n = P_j^{n-1} - \frac{\Delta t}{2\Delta x}\left((\mu P)_{j+1}^{n-1} - (\mu P)_{j-1}^{n-1}\right) + \frac{\Delta t}{\Delta x^2}\left((DP)_{j+1}^{n-1} + (DP)_{j-1}^{n-1} - 2(DP)_j^{n-1}\right) \tag{4}$$

It is important to note that the forward Euler method, *Equation 4*, is an approximation method. Since the probability distribution is iteratively propagated from early to later times, any errors of the solution accumulate over the iterative process. Moreover, the forward Euler method is prone to instability. Assuming small $\Delta x$ and $\Delta t$ (i.e. much less than 1), $\frac{\Delta t}{\Delta x^2}$ is much larger than $\frac{\Delta t}{2\Delta x}$ and thus the third term on the right side of *Equation 4* should in general be much larger than the second term. If $\Delta x$ and $\Delta t$ are such that the third term is large for any $n$ and $j$, then $P_j^n$ will not be properly computed in *Equation 4*, and such errors will propagate to the whole space over time. The forward Euler method thus has a stability criteria:

$$D\frac{\Delta t}{\Delta x^2} < \frac{1}{2} \tag{5}$$

*Equation 5* poses a strict constraint on the step-sizes: while $\Delta x$ and $\Delta t$ should both be small to minimize the error due to discretization, decreasing $\Delta x$ demands a much stronger decrease in $\Delta t$, resulting in a much longer execution time of the method. This can be seen in our simulations shown in *Figure 4*.

The backward Euler and Crank-Nicolson methods largely circumvent the aforementioned problems. In the backward Euler method, the right side of *Equation 2* is expressed at the current time step $n$:

$$\frac{P_j^n - P_j^{n-1}}{\Delta t} = -\frac{(\mu P)_{j+1}^n - (\mu P)_{j-1}^n}{2\Delta x} + \frac{(DP)_{j+1}^n + (DP)_{j-1}^n - 2(DP)_j^n}{\Delta x^2}, \tag{6}$$

In the Crank-Nicolson method, half of the right side of *Equation 2*) is expressed at the current time step $n$, and half at the previous time step $n-1$:

$$\frac{P_j^n - P_j^{n-1}}{\Delta t} = 0.5\left(-\frac{(\mu P)_{j+1}^{n-1} - (\mu P)_{j-1}^{n-1}}{2\Delta x} + \frac{(DP)_{j+1}^{n-1} + (DP)_{j-1}^{n-1} - 2(DP)_j^{n-1}}{\Delta x^2}\right)$$
$$+ 0.5\left(-\frac{(\mu P)_{j+1}^{n} - (\mu P)_{j-1}^{n}}{2\Delta x} + \frac{(DP)_{j+1}^{n} + (DP)_{j-1}^{n} - 2(DP)_j^{n}}{\Delta x^2}\right), \tag{7}$$

With multiple terms at the current time-step, *Equation 6* and *Equation 7* cannot be directly solved. Instead, the equations across all spatial grids can be summarized in matrix form:

$$(\mathbb{1} + (1-\alpha)\mathbb{M})\vec{P}^n = (\mathbb{1} - \alpha\mathbb{M})\vec{P}^{n-1}$$

$$\text{for } \vec{P}^n = \begin{pmatrix} P_{x=-B+\Delta x}^n \\ \vdots \\ P_{x=B-\Delta x}^n \end{pmatrix}, \mathbb{M} = \begin{pmatrix} 2\nu & -\nu+\chi/2 & 0 & 0 & 0 \\ -\nu-\chi/2 & 2\nu & -\nu+\chi/2 & 0 & 0 \\ 0 & -\nu-\chi/2 & \ddots & \ddots & 0 \\ 0 & 0 & \ddots & \ddots & -\nu+\chi/2 \\ 0 & 0 & 0 & -\nu-\chi/2 & 2\nu \end{pmatrix}, \tag{8}$$

Here, $\nu = D\Delta t/\Delta x^2$ and $\chi = \mu\Delta t/\Delta x$. In this formulation, $\alpha = 1$ for the forward Euler method, $\alpha = 0$ for the backward Euler method, and $\alpha = 0.5$ for the Crank-Nicolson method. Since $\mathbb{M}$ is a tridiagonal matrix, it can be inverted from the left side to the right side of the equation in an efficient manner. Practically, this allows the backward Euler and Crank-Nicolson method to rapidly generate the discretized probability distribution function inside the boundaries at each time-step.

Finally, the probability of decision formation at each time step is computed as the outward flux from the outermost grids $\pm B \mp \Delta x$ (to the hypothetical grids at $\pm B$):

$$\begin{aligned} \Delta p_{correct}^n &= (-\nu - \chi/2)|_{x=B-\Delta x}^n \, p_{x=B-\Delta x}^n \\ \Delta p_{incorrect}^n &= (-\nu + \chi/2)|_{x=-B+\Delta x}^n \, p_{x=-B+\Delta x}^n \end{aligned} \tag{9}$$

This provides the correct and incorrect reaction time distributions, as well as the total correct ($\sum_n \Delta p_{correct}^n$) and incorrect ($\sum_n \Delta p_{incorrect}^n$) probability.

Unlike the forward-Euler method which iteratively propagates the solution and the approximation error in a feedforward manner, the backward Euler and Crank-Nicolson methods solve for the matrix equation of the probability distribution evolution, and are less susceptible to error being propagated. In addition, the two methods do not have any stability criteria which constrain the step size. Contrasting the two methods, the backward Euler method has larger truncation error ($O(\Delta x, \Delta t^2)$) than the Crank-Nicolson method ($O(\Delta x^2, \Delta t^2)$) (*Table 1*), while the Crank-Nicolson method generally has twice the execution time of the backward Euler method, assuming the matrix construction is the slowest step of the algorithms. Crucially, the solution from the Crank-Nicolson method tends to be susceptible to oscillations over iterations, especially with varying bounds (see below). PyDDM automatically chooses the best solver for any given model (*Table 1*).

## Fokker-Planck formalism with time-varying bounds

The previous section considered the Fokker-Planck equation of the GDDM assuming fixed bounds. The formalism can be adapted to accommodate time-varying bounds. This is more generally studied as a free boundary problem, with a typical example including the Stefan problem (*Meyer, 1978*; *Li, 1997*).

In particular, consider upper and lower bounds $B_n = B(n\Delta t, \dots)$ and $-B_n = -B(n\Delta t, \dots)$ at an arbitrary time $t = n\Delta t$. If $B_n$ is exactly on a grid point, the previous results apply. Alternatively, if $B_n$ is in between two grid points $(x_j, x_{j+1})$, the solution is approximated by a linearly weighted summation of the solutions in two grid systems: an inner grid system $\{x_k | k = -j, \dots, j\}$ and an outer grid system $\{x_k | k = -j-1, \dots, j+1\}$.

Define $\vec{P}_{in/out}^n$ to be the probability distribution function at time $t = n\Delta t$ over the inner and outer grid systems, respectively. In the backward Euler method ($\alpha = 0$ in *Equation 8*), $\vec{P}_{in/out}^n$ can be computed by propagating from the actual probability distribution function at the previous step $\vec{P}^{n-1}$:

$$(\mathbb{1} + \mathbb{M}_{in/out})\vec{P}^n_{in/out} = \vec{P}^{n-1}, \tag{10}$$

where $\mathbb{M}_{in/out}$ are as in **Equation 8**, but defined over the domain of inner/outer grids respectively.
Furthermore, define

$$\begin{aligned}
w_{out} &= (B_n - x_j)/\Delta x \\
w_{in} &= (x_{j+1} - B_n)/\Delta x
\end{aligned} \tag{11}$$

as the weights of the linear approximation of the solution to that of the outer and inner grid systems.
The resulting probability distribution function and the probabilities of decision formation at step $n$
are:

$$\begin{aligned}
\vec{P}^n &= w_{out}\vec{P}^n_{out} + w_{in}\vec{P}^n_{in} \\
\Delta p^n_{correct} &= w_{out}(-\nu + \chi/2)|^n_{x=x_{j+1}} P^n_{j+1,out} + w_{in}(-\nu + \chi/2)|^n_{x=x_j} P^n_{j,in} \\
\Delta p^n_{incorrect} &= w_{out}(-\nu - \chi/2)|^n_{x=x_{-j-1}} P^n_{-j-1,out} + w_{in}(-\nu - \chi/2)|^n_{x=x_{-j}} P^n_{-j,in}
\end{aligned} \tag{12}$$

In the literature of the DDM in cognitive psychology and neuroscience, collapsing bounds are the
most common form of time-varying bounds considered. As bounds collapse, it is possible that the
outermost grids at the previous step (with non-zero probability densities) become out-of-bound at
the current step. Based on the side they become out of bound, such probabilities are added to the
correct and incorrect probabilities respectively (and multiplied by $w_{out/in}$ for each of the outer/inner
grid system). Nevertheless, increasing bounds are also supported. The algorithm for the Crank-Nic-
olson scheme of time-varying bounds often generates oscillatory solutions, and is thus not described
here or implemented in PyDDM.

## Empirical datasets

As testbeds for fitting different models, we used two empirical datasets of choices and RTs during
perceptual decision-making tasks: one from *Roitman and Shadlen, 2002*, one from *Evans and Haw-
kins, 2019*. These references contain full details of the tasks and datasets. For the dataset of
*Roitman and Shadlen, 2002*, we used trials from the 'reaction time' task variant, for both monkeys.
For the dataset of *Evans and Hawkins, 2019*, we used trials from the task variant with no feedback
delay.

In brief, these two publicly available datasets were selected for the following reasons. Both use
random dot motion discrimination tasks, which facilitate comparison of behavior and fitting with the
same GDDM. These two datasets span species, with behavior from monkeys (*Roitman and Shadlen,
2002*) and human subjects (*Evans and Hawkins, 2019*). Furthermore, both datasets have been ana-
lyzed in prior work fitting DDMs with collapsing bounds. Specifically, *Hawkins et al., 2015a* gener-
ally found that the dataset of *Roitman and Shadlen, 2002* was better fit with collapsing bounds
than with constant bounds, in contrast with typical human datasets which are well fit with constant
bounds. In line with those findings, *Evans and Hawkins, 2019* showed that behavior in a task variant
with no feedback delay yielded behavior which was more representative of human datasets and was
well fit by a DDM with decision bounds with no or little collapse. These two datasets therefore pro-
vided useful testbeds for GDDM fitting with distinct a priori predictions about the utility of collaps-
ing bounds in a GDDM.

## Compared models

The 6-parameter GDDM fit by PyDDM is parameterized by

$$\begin{aligned}
\mu(x, t, \ldots) &= \mu_0(C - C_{\max})^\alpha - \ell x \\
\sigma(x, t, \ldots) &= 1 \\
X_0(x, \ldots) &= \begin{cases} 1, & x = 0 \\ 0, & \text{otherwise} \end{cases} \\
B(t, \ldots) &= B_0 e^{-t/\tau} \\
p^*_i(t) &= 0.95 p_i(t - t_{nd}) + 0.025
\end{aligned} \tag{13}$$

where fittable parameters are drift rate $\mu_0$, coherence nonlinearity $\alpha$, leak constant $\ell$, initial bound

height $B_0$, bound collapse rate $\tau$, and non-decision time $t_{nd}$. Additionally, $C_{\max}$ is the maximum coherence in the experiment, which is fixed at 0.512 for the *Roitman and Shadlen, 2002* dataset and 0.4 for the *Evans and Hawkins, 2019* dataset.

The three-parameter DDM fit by PyDDM is a subset of the GDDM framework parameterized by

$$
\begin{aligned}
\mu(x,t,\dots) &= \mu_0 C \\
\sigma(x,t,\dots) &= 1 \\
X_0(x,\dots) &= \begin{cases} 1, & x=0 \\ 0, & \text{otherwise} \end{cases} \\
B(t,\dots) &= B_0 \\
p_i^*(t) &= 0.95 p_i(t - t_{nd}) + 0.025
\end{aligned}
\tag{14}
$$

where $C$ is coherence and fittable parameters are $\mu_0$, $B_0$, and $t_{nd}$.

The 11-parameter 'full DDM' fit by HDDM does not fall into the GDDM framework because the across-trial variability in drift rate $\mu(x,t,\dots)$ is a random variable. Thus, the 'full DDM' is parameterized by

$$
\begin{aligned}
\mu(x,t,\dots) &\sim \mathcal{N}(\mu_j, s_v^2) \\
\sigma(x,t,\dots) &= 1 \\
X_0(x,\dots) &= \begin{cases} \frac{1}{2s_z+1}, & -s_z \le x \le s_z \\ 0, & \text{otherwise} \end{cases} \\
B(t,\dots) &= B_0 \\
p_i^*(t) &= \frac{1}{2s_T+1} \sum_{k\in[-s_T,s_T]} p_i(t - t_{nd} - k)
\end{aligned}
\tag{15}
$$

where $\mathcal{N}$ is a normal distribution, and fittable parameters are six independent drift rates $\mu_j$ for each coherence level $j$, non-decision time $t_{nd}$, and bounds $B_0$, as well as variability in drift rate $s_v$, starting position $s_z$, and non-decision time $s_T$. It was fit an outlier probability of $\text{p\_outlier} = 0.05$.

The 8-parameter DDM fit by HDDM is a subset of the GDDM framework parameterized by

$$
\begin{aligned}
\mu(x,t,\dots) &= \mu_j \\
\sigma(x,t,\dots) &= 1 \\
X_0(x,\dots) &= \begin{cases} 1, & x=0 \\ 0, & \text{otherwise} \end{cases} \\
B(t,\dots) &= B_0 \\
p_i^*(t) &= p_i(t - t_{nd})
\end{aligned}
\tag{16}
$$

where fittable parameters are $B_0$, $t_{nd}$, and six independent drift rates $\mu_j$, one for each coherence level $j$. It was fit an outlier probability of $\text{p\_outlier} = 0.05$.

The 18-parameter DDM fit by EZ-Diffusion is a subset of the GDDM framework parameterized by

$$
\begin{aligned}
\mu(x,t,\dots) &= \mu_j \\
\sigma(x,t,\dots) &= 1 \\
X_0(x,\dots) &= \begin{cases} 1, & x=0 \\ 0, & \text{otherwise} \end{cases} \\
B(t,\dots) &= B_j \\
p_i^*(t) &= p_i(t - t_{nd}^j)
\end{aligned}
\tag{17}
$$

where fittable parameters are independent drift rates $\mu_j$, bounds $B_j$, and non-decision times $t_{nd}^j$, for each of the six coherence level $j$.

All simulations were fit separately for each monkey. In the human fits, data were pooled across all subjects for the 0 s delay condition only.

## Execution time and accuracy analysis

To analyze the execution time vs error tradeoff, we benchmarked a DDM which fits into the GDDM framework, defined by:

$$\begin{aligned}
\mu(x,t,\ldots) &= 2 \\
\sigma(x,t,\ldots) &= 1.5 \\
X_0(x,\ldots) &= \begin{cases} 1, & x=0 \\ 0, & \text{otherwise} \end{cases} \\
B(t,\ldots) &= 1 \\
p_i^*(t) &= p_i(t)
\end{aligned} \tag{18}$$

The DDM was chosen because it permits closed-form solutions, providing a comparison for benchmarking simulation accuracy. Performance was measured on a Lenovo T480 computer with an Intel M540 i7-8650U 4.2 GHz CPU with hyperthreading disabled. Verification and parallelization were disabled for these evaluations. The simulated time was 2 s.

In order to examine combinations of $\Delta t$ and either $\Delta x$ or $N$ which simultaneously minimize both execution time and error, we plot the product of the MSE and execution time in seconds. This product should be interpreted as a heuristic which encompasses both execution time and accuracy rather than a fundamentally important quantity in and of itself.

## Model for parallel performance

To analyze PyDDM's parallel performance, we benchmarked a DDM which fits into the GDDM framework, defined by:

$$\begin{aligned}
\mu(x,t,\ldots) &= 2C \\
\sigma(x,t,\ldots) &= 1 \\
X_0(x,\ldots) &= \begin{cases} 1, & x=0 \\ 0, & \text{otherwise} \end{cases} \\
B(t,\ldots) &= 1 \\
p_i^*(t) &= p_i(t)
\end{aligned} \tag{19}$$

where the model was simulated using backward Euler for 64 coherence values $C$ uniformly spaced from 0 to 0.63. Because PyDDM parallelizes functions by utilizing an embarrassingly parallel approach over experimental conditions, we included many such conditions to enable benchmarks which reflect the simulations instead of the specifics of the model. Simulations were performed on a high performance computing cluster using a single node with 28 Xeon E5-2660v4 cores. Verification was disabled for these simulations.

## Validity of results

It is important that simulation results are valid and reliable. Results could be inaccurate either due to bugs in user-specified models or in PyDDM itself. Due to the ability to easily construct user-defined models, PyDDM takes this issue very seriously. In addition to utilizing standard software engineering practices such as unit and integration testing with continuous integration (*Beck Andres, 2004*), PyDDM is the first neuroscience software to our knowledge which incorporates software verification methods for ensuring program correctness (*Ghezzi et al., 2002*). It does so using a library which checks for invalid inputs and outputs within the code, and also checks for invalid inputs and outputs in user-defined models which do not explicitly incorporate the library (*Shinn, 2020*). Invalid inputs and outputs include anything which invalidates the assumptions of the GDDM, such as initial conditions outside of the bounds, noise levels less than zero, or probability distributions which do not sum to 1. Such checking helps ensure that results generated by PyDDM are not due to a configuration or programming mistake, which is especially important because user-specified models logic can quickly become complicated for intricate experimental designs. This, combined with the ease of reproducibility and code reuse afforded by open source software in general, means that high validity and reliability are key advantages of PyDDM.

## Acknowledgements

This research was supported by NIH grant R01MH112746 (JDM), the Gruber Foundation (MS), and NSERC (NHL). We thank Robert Yang for his contribution to the analytic solver, and Daeyeol Lee

and Hyojung Seo for helpful discussions. We also acknowledge the authors of *Roitman and Shadlen, 2002* and *Evans and Hawkins, 2019* for making their datasets publicly available.

# Additional information

## Funding

| Funder | Grant reference number | Author |
|---|---|---|
| National Institute of Mental Health | R01MH112746 | John D Murray |
| Gruber Foundation | | Maxwell Shinn |
| NSERC | PGSD2-502866-2017 | Norman H Lam |

The funders had no role in study design, data collection and interpretation, or the decision to submit the work for publication.

## Author contributions

Maxwell Shinn, Norman H Lam, Conceptualization, Software, Formal analysis, Visualization, Methodology, Writing - original draft, Writing - review and editing; John D Murray, Conceptualization, Supervision, Visualization, Methodology, Writing - original draft, Project administration, Writing - review and editing

## Author ORCIDs

Maxwell Shinn (ID) https://orcid.org/0000-0002-7424-4230
Norman H Lam (ID) https://orcid.org/0000-0001-5817-6680
John D Murray (ID) https://orcid.org/0000-0003-4115-8181

## Decision letter and Author response

Decision letter https://doi.org/10.7554/eLife.56938.sa1
Author response https://doi.org/10.7554/eLife.56938.sa2

# Additional files

## Supplementary files

• Transparent reporting form

## Data availability

The two analyzed datasets, which have been previously published, are both is publicly available for download: 1. Roitman & Shadlen, 2002, J Neurosci: https://shadlenlab.columbia.edu/resources/RoitmanDataCode.html 2. Evans & Hawkins, 2019, Cognition: https://osf.io/2vnam/.

The following previously published datasets were used:

| Author(s) | Year | Dataset title | Dataset URL | Database and Identifier |
|---|---|---|---|---|
| Roitman JD, Shadlen MN | 2002 | Data from: Roitman and Shadlen (2002) | https://shadlenlab.columbia.edu/resources/RoitmanDataCode.html | Shadlen Lab website, RoitmanDataCode |
| Evans NJ, Hawkins GE | 2019 | Data from: Evans and Hawkins (2019) | https://osf.io/2vnam/ | Open Science Framework, 2vnam |

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

## Appendix 1

## Implementing and fitting models in PyDDM

Briefly, we describe the process of building and fitting a model in PyDDM. For more detailed information, see the quickstart guide in the documentation at https://pyddm.readthedocs.io/en/latest/quickstart.html. A familiarity with Python and with Python classes is assumed.

PyDDM is based on a modular architecture, organized around the principles of the GDDM. There are five components, specified using Python classes, which correspond to the five functions listed in section 'GDDM description':

- Drift: the drift rate. May depend on time, decision variable position, task conditions, and fittable parameters. Defaults to 0.
- Noise: the standard deviation of diffusion process. May depend on time, decision variable position, fittable parameters, and task conditions. Defaults to 1.
- Bound: The integration bounds (+B and -B). May depend on time, fittable parameters, and task conditions. Defaults to 1.
- IC: The initial conditions at time t = 0, in terms of a probability density function over decision variable position. May depend on fittable parameters or task conditions. Defaults to a probability density function of a Kronecker delta function centered at $x = 0$.
- Overlay: Post-factum modifications to the histogram, for instance to create a mixture model or add a non-decision time. May depend on fittable parameters or task conditions. Defaults to an identity function.

These model components do not specify a single value for these parameters, but rather specify a function through which these values can be computed. Each may take any number of parameters and utilize any number of task variables. Most models will not implement all of these components from scratch, but will use the default component or a built-in component for most of them. For example, the GDDM in *Figure 2* (*Equation 13*) used the default 'Noise' and 'IC' components, built-in Bounds and Overlay components, and a customized 'Drift' component. A full list of built-in components, as well as a library of example components, can be found in the online documentation.

Each of these model components may depend on parameters; for example, the 'Bound' component requires the parameters $B_0$ and $\tau$. These parameters must be specified when the class for the model component is instantiated, for example, 'BoundCollapsingExponential' is the class for exponential collapsing bounds used in the GDDM in *Figure 2* (*Equation 13*). It can be instantiated (for $B_0 = 2$ and $\tau = 3$) as

```
BoundCollapsingExponential(B0 = 2, tau = 3)
```

Of course, we would often like to fit these parameters to data, and thus, we cannot specify their exact values when instantiating a component. For these situations, we use an instance of a 'Fittable' class as a placeholder. Minimum and maximum values must be specified for bounded fitting algorithms, such as differential evolution. For example, we could fit both of these parameters using

```
BoundCollapsingExponential(B0 = Fittable(minval = 0.5, maxval = 3),
tau = Fittable(minval = 0.1, maxval = 10))
```

Components which are not built-in to PyDDM may be specified by defining a class which gives four pieces of information: the component's name, the parameters it depends on, the task conditions it depends on, and a function to compute its value. For example, here is 'DriftCoherenceLeak', the 'Drift' class used in our GDDM in *Figure 2* (*Equation 13*):

```
class DriftCoherenceLeak (ddm.models.Drift):
  name = "Leaky drift depends nonlinearly on coherence"
  # Parameters we want to include in the model
  required_parameters = ["driftcoh", "leak", "power", "maxcoh"]
  # Task parameters/conditions. Should be the same name as in the sample.
  required_conditions = ["coh"]
  # The get_drift function is used to compute the instantaneous value of drift.
  def get_drift(self, x, conditions, **kwargs):
    return self.driftcoh * (conditions["coh"]/self.maxcoh)**self.power + self.leak
* x
```

Once we have determined all of the components of our model, we may tie them together using an instance of the 'Model' class. When creating an instance of the 'Model' class, we must also define our timestep 'dt', grid spacing 'dx', and simulation duration 'T_dur'. Shown below is the GDDM in *Figure 2* (*Equation 13*):

```
m=Model(name="Roitman-Shadlen GDDM",
    drift = DriftCoherenceLeak(driftcoh = Fittable(minval = 0, maxval = 20),
                    leak = Fittable(minval=-10, maxval = 10)),
    # Do not specify noise=… since we want the default
    # Do not specify IC=… since we want the default
    bound = BoundCollapsingExponential(B = Fittable(minval = 0.5, maxval = 3),
                        tau = Fittable(minval = 0.0001, maxval = 5)),
    # OverlayChain strings together multiple overlays.
    Overlay = OverlayChain(overlays=[
            OverlayNonDecision(nondectime = Fittable(minval = 0, maxval = 0.4)),
            OverlayUniformMixture(umixturecoef = 0.05)]),
    dx = 0.001, dt = 0.001, T_dur = 2)
```

If the model 'm' includes at least one component with a 'Fittable' instance, we must first fit the model to data by calling the 'fit_adjust_model' function, passing our model and dataset as arguments. By default, it fits using differential evolution (*Storn and Price, 1997*), however Nelder-Mead (*Nelder and Mead, 1965*), BFGS (*Nocedal and Wright, 2006*), basin hopping (*Wales and Doye, 1997*), and hill climbing (*Luke, 2013*) are also available. Fitting a model in parallel using *n* cores of a CPU is as simple as calling the function 'set_N_cpus(*n*)' before fitting the model. Once we load our sample 'samp' using one of a number of convenience functions for loading data, we may fit the model:

```
set_N_cpus(4)
fit_adjust_model(m, sample = samp)
```

After fitting the model to data, the 'Fittable' instances are automatically replaced by their fitted values, so the model can be directly simulated using 'm.solve()'. If any of the model components depend on task conditions, these must be passed as a dictionary to the solver; for example, since the 'DriftCoherenceLeak' component depends on 'coh' (the trial's coherence), we must instead call

```
m.solve(conditions={'coh': 0.064})
```

to simulate with a coherence of 0.064. Alternatively, the function 'solve_partial_conditions' can simulate across many task conditions simultaneously. Both the functions 'm.solve' and 'solve_partial_conditions' will return an instance of the 'Solution' class, which has methods for obtaining the correct and error RT distributions as well as useful summary statistics.

