## [Decision Letter]

**Acceptance summary:**

Drift diffusion models are widely used in psychology and neuroscience research. However, diffusion model applications are often constraint by the available software rather than the scientific question that ought to be asked. Thus, having a model and associated software package that provides more flexible model fits will be of great value for the scientific community.

**Decision letter after peer review:**

Thank you for submitting your article "A flexible framework for simulating and fitting generalized drift-diffusion models" for consideration by *eLife*. Your article has been reviewed by Joshua Gold as the Senior Editor, a Reviewing Editor, and three reviewers. The following individuals involved in review of your submission have agreed to reveal their identity: Long Ding (Reviewer #1); Eric-Jan Wagenmakers (Reviewer #2).

The reviewers have discussed the reviews with one another and the Reviewing Editor has drafted this decision to help you prepare a revised submission.

We would like to draw your attention to changes in our revision policy that we have made in response to COVID-19 (https://elifesciences.org/articles/57162). Specifically, we are asking editors to accept without delay manuscripts, like yours, that they judge can stand as *eLife* papers without additional data, even if they feel that they would make the manuscript stronger. Thus, the revisions requested below only address clarity and presentation.

Summary:

Reviewers were generally impressed with this work as it presents a substantial step forward, providing a much-needed, flexible, and efficient way to fit choice and reaction time data common to many decision behaviors. Reviewers mostly made a number of suggestions for how to improve the manuscript. However, they also identified two essential issues that would need to be adequately addressed in a revised version. Specifically, it would be important to apply the models to a different data set and to consider additional criteria for model evaluation.

Essential revisions:

1) As explained in reviewer #2's major comment 2, the RT distribution in the Roitman and Shadlen data set is not very representative. Reviewers agree that the authors should include an addition, more representative data set for the current paper. In particular, reviewers suggested the following data set: Wagenmakers et al., (2008). These data are available at http://www.ejwagenmakers.com/Code/2008/LexDecData.zip

2) As detailed in reviewer #2's major comment 3, and reviewer #3 major comment 3, given the flexibility of the model, it is unclear whether goodness-of-fit is the only or even the most appropriate criteria to evaluate different models.

Reviewer #1:

This work provides a much-needed, flexible, and efficient way to fit choice and reaction time data common to many decision behaviors.

The detailed methods in the section "Fokker-Plank formalism of the GDDM" are outside my expertise. But the superior performance of the GDDM compared to other packages, both in accuracy and execution time, is very convincing and rigorously demonstrated. The writing is also very clear. I expect that the software package will be widely used in the field. This is a rare instance when I support publication as is.

Reviewer #2:

In general I am truly impressed with this work. It present a substantial step forward, and I strongly believe a revision should be published. At the same time, I also believe the manuscript can be greatly strengthened along the lines provided below.

Essential revisions:

1) "Fifth, single trial parameter variability is supported within the GDDM framework for both starting point and non-decision time."

Initially, I was not sure what the authors mean with "single trial parameter variability". Do they mean "across-trial variability"? And if so, does their methodology not allow for across-trial variability in drift rate, as the full DDM does? If the GDDM indeed does not allow such variability, this warrants explicit mention and an extensive discussion.

Then when I read the Discussion section the cat comes out of the bag: "While the GDDM framework supports individual trial variability in non-decision time and starting position according to any arbitrary distribution, it does not support drift rate variability. As a result, the GDDM framework does not support the entire "full DDM"."

But this is a severe limitation, one that needs to be acknowledged and emphasized from the start. Variability in drift rate is a highly plausible mechanism that has been an integral part of the DDM since 1978. Without this mechanism to account for (fairly ubiquitous) slow errors, the model will have to turn to other mechanism to accomplish this, such as moving-in bounds. The authors mention "However, the GDDM framework is broad enough to support an infinite number of extensions which create the same RT features which originally motivated inclusion of drift rate variability." But the goal of the modeling is not to fit the data well; the actual goal is to decompose the observed data into meaningful psychological processes, and a process of moving-in bounds is fundamentally different from drift-rate variability. I assume that the drift rate variability frustrates the mathematical work? The reader needs to know the reason why the model cannot handle drift rate variability.

Staying on the same point, the authors then point out that "Indeed, in Figure 2, our 5-parameter GDDM provided a better fit to the asymmetry between correct and error RT distributions than the 11-parameter "full DDM"." True, but since these data are highly unrepresentative (see the next point below), this can just as well be used as an argument *against* the GDDM: apparently it is able to fit just about any pattern of data, even patterns that are highly unrepresentative.

2) The demonstration where the models are fit to the data is (deeply) problematic, because the monkey data are simply not representative. As can be seen from the grey histograms in Figure 2, the "regular" DDMs predict a long-tailed RT distribution, which is however not apparent from the data. This then creates ambiguity: yes, one can show that the existing models do not fit well, but they are fit to unrepresentative data. If these data patterns were regularly observed, the DDM would never have been proposed in the first place. It would be much more compelling to fit the models to any of the large benchmark data sets that are publicly available – these data sets all show a pronounced right skew.

3) As the progenitor of the much-maligned "EZ" model, I guess I am obliged to protest a little. Yes, of course EZ is not flexible, and does not have all the bells and whistles. But that is kind of the point of a simple model. There are some demonstrations that (contrary to my own expectations even) EZ actually does better than the full diffusion model when applied to sparse data. I am not talking about goodness-of-fit, but about recovering the ground truth in a simulation study (e.g., figuring out which parameters are the same and which differ). For sparse data, a simple model may work better. One reference:

van Ravenzwaaij, Donkin and Vandekerckhove, (2017).

Reviewer #3:

The manuscript introduces a generalized drift-diffusion model, together with the modular software package PyDDM to fit it. The generalized drift-diffusion model is a set of accumulation-to-bound models that feature highly flexible components, such as time- and location-varying drifts, variable starting points, time-varying decision boundaries, etc. The manuscript describes how to compute first-passage time distributions for this set of models by numerically and efficiently solving the Fokker-Plank Equation (FPE), of which PyDDM provides an implementation, together with a set of optimizers that can be used to adjust the model's parameters to behavioral data.

Having a model and associated software package that provides more flexible model fits is very welcome and timely, as diffusion model applications are frequently constraint by the available software rather than the scientific question that ought to be asked. PyDDM promises to provide such a package, and appears to be sufficiently well document to ease uptake by the community. Thus, it would be a welcome addition to the currently available diffusion model fitting packages.

Some concerns should be addressed before the manuscript can be published:

1) The manuscript frequently describes the numerical solution of the FPE a "simulation", starting with the title, the introduction, the main text, etc. Commonly, solving the FPE is not considered a simulation, such that describing it as such is potentially confusing misnomer – in particular as there exist simulation-based "likelihood-free" methods to fit models solely based on simulating them. First, I urge the authors to not call solving FPEs a "simulation", but instead describe their approach as numerically propagating the probability mass, which can intuitively be thought of as performing an infinite number of simulations. Second, the authors should acknowledge likelihood-free fitting methods, like ABC (see, for example, Holmes and Trueblood, 2018 and related) to fit diffusion models – subsection “Methods for simulating the DDM and GDDM” would be a good starting point.

2) The precision and the clarity of the manuscript can be improved in many instances:

- It should be made clear that GDDMs only support Markovian processes + some additional post-processing of the computed first-passage time distributions. This is, for example, the reason why they don't support continuously varying drift rates, but this isn't at all clear from the manuscript. An additional (current) limitation is that the implementation currently only supports boundaries that vary symmetrically around zero, even though this appears to be a limitation of the provided implementation rather than a fundamental limit to the chosen approach.

- The manuscript frequently claims that the solution is "continuous" (e.g., subsection “Methods for simulating the DDM and GDDM”, subsection “Software package for GDDM fitting”) in time, even though they only provide time-discretized numerical results.

- It isn't sufficiently clear from the manuscript that all provided FPE solutions are approximate, as they discretize time and space. Currently, the manuscript only highlights the approximate nature of the forward Euler scheme (subsection “Execution time vs error tradeoff”). Indeed, it has worse numerical properties, but that doesn't make the implicit schemes non-approximate.

3) The manuscript highlights a better fit (i.e., higher likelihood, better BIC, etc.) of a DDM with time-varying boundaries and a leak, and uses this fit (of a single dataset) to re-iterate throughout the manuscript that a better model fit with fewer parameters is always more desirable (e.g., subsection “Software package for GDDM fitting”). This mis-characterizes the modeling exercise, in which the structure and presence of different components of DDMs might be as important as the quality of the fit.

4) The authors highlight three approaches to "simulate" DDMs and GDDMs, but seem to be missing a fourth based on solving Volterra integral equation, as described by Smith, (2000) and implemented by the 'dm' package by Drugowitsch et al., – which is fairly flexible, but unfortunately only provides computing the first-passage times, but no model fitting code.

---

## [Author Response]

Essential revisions:1) As explained in reviewer #2's major comment 2, the RT distribution in the Roitman and Shadlen data set is not very representative. Reviewers agree that the authors should include an addition, more representative data set for the current paper. In particular, reviewers suggested the following data set: Wagenmakers et al., (2008). These data are available at http://www.ejwagenmakers.com/Code/2008/LexDecData.zip

We thank the reviewers for this helpful suggestion, which we think will better contextualize the role of GDDMs in working with empirical datasets. We had utilized the Roitman-Shadlen dataset as an example testbed to demonstrate how GDDMs could be fit to empirical dataset to test hypothesized mechanisms (there, collapsing bounds and leaky integration, as motivated by prior studies). We agree that RT distributions in this dataset are not representative of many experiments from human subjects, and therefore that also fitting models to a representative human dataset is an important addition to the paper.

We thank reviewer 2 for the suggestion of the publicly available dataset from Wagenmakers et al., (2008a). However, after surveying other publicly available datasets, we decided that it is not most appropriate for this analysis, especially to serve as a comparison to Roitman and Shadlen, (2002). The mechanisms we focused on in this manuscript are those related to integration of evidence extended over time. In the dataset of Wagenmakers et al., (2008a), the task was to determine whether a string of characters formed a word or not, and collapsing bounds and leaky integration do not have clear interpretations as cognitive strategies for this task paradigm (see our response to Essential revision major point 2).

To address the reviewer’s concern, we sought to include a more representative dataset from human subjects which would also be more comparable in task design to the monkey experiment of Roitman and Shadlen, (2002). Here we were primarily inspired by the recent publication of Hawkins et al., (2015a) (“Revisiting the evidence for collapsing boundaries and urgency signals in perceptual decision-making”), which analyzed multiple studies from monkeys and human subjects performing perceptual decision-making tasks and specifically investigated evidence for collapsing bounds. Overall, they found that most monkey datasets better fit by collapsing bounds than fixed bounds, but that most human datasets were better fit by fixed bounds. Hawkins et al., (2015a) suggested these differences may arise from task design and training differences, as well as species differences.

In our revised manuscript, we now use publicly available data from the study of Evans and Hawkins, (2019) (“When humans behave like monkeys: Feedback delays and extensive practice increase the efficiency of speeded decisions”), which tested the Roitman-Shadlen task paradigm in human subjects, and manipulated feedback timing and amount of practice. The dominant effect they found was in feedback timing: having zero delay between response and feedback yielded RTs that were best fit by no or little collapse of bounds (similar to the human studies analyzed by Hawkins et al., (2015a)); whereas a 1-s delay in feedback (as in Roitman and Shadlen, (2002)) yielded RTs that were best fit by collapsing bounds (similar to the monkey studies analyzed by Hawkins et al. (2015a)). We therefore selected for our GDDM analysis the zero-delay trials from Evans and Hawkins, (2019), which the authors found are more representative of human datasets.

Using the two publicly available datasets of (i) Roitman and Shadlen, (2002) and (ii) Evans and Hawkins, (2019) as testbeds for our GDDM framework has several advantages. They span monkey and human studies. They use very similar random dot motion perceptual decision-making task paradigms. They allow us to test the same GDDM (i.e., with collapsing bounds, leaky integration, and input nonlinearity) across datasets. Finally, they provide benchmarks for comparison to PyDDM through different model-fitting methodologies: Hawkins et al., (2015a) found that the Roitman and Shadlen, (2002) dataset was better fit with collapsing bounds, whereas Evans and Hawkins, (2019) found that their dataset was best fit with no or little collapse.

We found that in the Evans-Hawkins dataset, the pre-specified GDDM did not perform better than the other models, but was more or less comparable. This contrast between the Roitman-Shadlen and EvansHawkins datasets nicely illustrates how our GDDM framework can be applied to test hypothesized mechanisms with empirical datasets, with example results that are consistent with prior literature. We now include a new Figure 2—figure supplement 1 on the Evans-Hawkins dataset, in parallel format to Figure 2.

We note that EZ-Diffusion performed especially poorly on the Evans-Hawkins dataset; in particular, the non-decision time was fit to be less than zero. We confirmed this was the case using two separate implementations of EZ-Diffusion: the javascript implementation, and the implementation built in to the HDDM Python package. The reason for this poor performance appears to be largely due to the inclusion of long RTs. We mirrored Evans and Hawkins, (2019) in using a cutoff of 7 seconds for the RT distribution. The high cutoff leads to a more skewed distribution with larger variance and a relatively smaller mean.

We introduced Figure 2—figure supplement 3 which describes the Evans-Hawkins dataset in a similar form as the Roitman-Shadlen dataset:

We added the following to describe this figure in the Results section:

"The RT distributions produced by monkeys may not be representative of those produced by humans (e.g., Hawkins et al., 2015a), which show a characteristic skewed distribution. […] This lack of improved fit by the GDDM suggests that the human subjects employed a cognitive strategy closer to the standard DDM, and that the specific mechanisms of this GDDM (collapsing bounds, leaky integration, and input nonlinearity) are not import for explaining psychophysical behavior in this dataset."

Additionally, we updated the Materials and methods section to include information about how we fit these datasets, as well as a new subsection briefly describing the two datasets and our rationales for their selection:

"As testbeds for fitting different models, we used two empirical datasets of choices and RTs during perceptual decision-making tasks: one from Roitman and Shadlen, (2002), one from Evans and Hawkins, (2019). These references contain full details of the tasks and datasets. For the dataset of Roitman and Shadlen, (2002), we used trials from the ‘‘reaction time’’ task variant, for both monkeys. For the dataset of Evans and Hawkins, (2019), we used trials from the task variant with no feedback delay."

"In brief, these two publicly available datasets were selected for the following reasons. […] These two datasets therefore provided useful testbeds for GDDM fitting with distinct *a priori* predictions about the utility of collapsing bounds in a GDDM."

2) As detailed in reviewer #2's Essential revision 3, and reviewer #3 Essential revision 3, given the flexibility of the model, it is unclear whether goodness-of-fit is the only or even the most appropriate criteria to evaluate different models.

We thank the reviewers for pointing out to us that our intention of the manuscript and the role of the GDDM were not sufficiently clear. Our intention appears to be in line with that of the reviewers. Our manuscript previously gave the impression that the best feature of the GDDM was its ability to fit datasets more accurately and with fewer parameters than other models. However, we stress that our goal is not to suggest that a GDDM (let alone the specific one demonstrated here) is generically the most appropriate model. Furthermore, we agree that the goal of modeling is not simply to fit data with the least error and/or fewest parameters. Our goal with this study is to provide a flexible, powerful, and accessible framework which (i) makes it easy to test new hypothesized cognitive mechanisms by fitting them to data, and (ii) to model task paradigms with time-varying evidence, which are difficult or impossible within the classic DDM framework. We used the Roitman-Shadlen dataset as an example testbed to demonstrate how hypothesized mechanisms can be fit to empirical data and how evidence for a mechanism can be informed by improvement in model fit. This is now further complemented by inclusion of the Evans-Hawkins dataset.

We have done the following to highlight these changes and clarify how GDDMs can be used in the scientific process. First, we have slightly changed the specific GDDM which is used in the context of the empirical datasets in section “GDDM mechanisms characterize empirical data”. Since our goal is to highlight the possibilities of using GDDMs in general, rather than one specific GDDM with collapsing bounds and leaky integration, we added another feature/parameter to the model which does not substantially improve fit but highlights another attractive feature of the GDDM: a fittable coherence nonlinearity. While this adds an extra parameter (the exponent of the nonlinearity), it allows us to demonstrate another practical and interesting feature of GDDMs.

We focus on this new description of the model within the Results section. The first paragraph of the subsection “GDDM mechanisms characterize empirical data” was changed to the following:

"The cognitive mechanisms by which evidence is used to reach a decision is still an open question. […] However, consistent with previous results (Hawkins et al., 2015a), the GDDM fit no better than the DDM or ‘‘full DDM’’ in the human dataset, suggesting that different cognitive strategies may be used in each case."

As a result of these changes, we change the subsection title to the following: "GDDM mechanisms characterize empirical data"

We have added the following paragraph to the Results section:

"For the example datasets considered above, the GDDM presented here improved the fit to the monkey data and provided a good fit to the human data. We caution that this result does not necessarily mean that this particular GDDM is appropriate for any particular future study. Models should not only provide a good fit to data, but also be parsimonious in form and represent a clear mechanistic interpretation (Vandekerckhove et al., 2015). The examples here demonstrate the utility of GDDMs for quantitatively evaluating cognitive mechanisms of interest through fitting models to empirical data."

We added the following paragraph to the Discussion section:

"PyDDM enables researchers to build very complex models of decision-making processes, but complex models are not always desirable. Highly complex models may be able to provide an excellent fit to the data, but unless they represent specific, plausible cognitive mechanisms, they may not be appropriate. PyDDM’s complexity can be used to examine a range of potential cognitive mechanisms which were previously difficult to test, and to probe a range of task paradigms."

Furthermore, we have removed the passages of the Roitman-Shadlen part of the Results section which imply that goodness of fit is more critical than the interpretability of the model over the goodness of fit.

Reviewer #2:Essential revisions:1) "Fifth, single trial parameter variability is supported within the GDDM framework for both starting point and non-decision time."Initially, I was not sure what the authors mean with "single trial parameter variability". Do they mean "across-trial variability"? And if so, does their methodology not allow for across-trial variability in drift rate, as the full DDM does? If the GDDM indeed does not allow such variability, this warrants explicit mention and an extensive discussion.Then when I read the Discussion section the cat comes out of the bag: "While the GDDM framework supports individual trial variability in non-decision time and starting position according to any arbitrary distribution, it does not support drift rate variability. As a result, the GDDM framework does not support the entire "full DDM"."But this is a severe limitation, one that needs to be acknowledged and emphasized from the start. Variability in drift rate is a highly plausible mechanism that has been an integral part of the DDM since 1978. Without this mechanism to account for (fairly ubiquitous) slow errors, the model will have to turn to other mechanism to accomplish this, such as moving-in bounds. The authors mention "However, the GDDM framework is broad enough to support an infinite number of extensions which create the same RT features which originally motivated inclusion of drift rate variability." But the goal of the modeling is not to fit the data well; the actual goal is to decompose the observed data into meaningful psychological processes, and a process of moving-in bounds is fundamentally different from drift-rate variability. I assume that the drift rate variability frustrates the mathematical work? The reader needs to know the reason why the model cannot handle drift rate variability.Staying on the same point, the authors then point out that "Indeed, in Figure 2, our 5-parameter GDDM provided a better fit to the asymmetry between correct and error RT distributions than the 11-parameter "full DDM"." True, but since these data are highly unrepresentative (see the next point below), this can just as well be used as an argument *against* the GDDM: apparently it is able to fit just about any pattern of data, even patterns that are highly unrepresentative.

We thank the reviewer for drawing our attention to the importance of supporting the entire “full DDM”. Fokker-Planck methods, in general, do not readily support true across-trial variability in drift rate, and therefore this constitutes a limitation of the Fokker-Planck approach. Each numerical solution of the Fokker-Planck equation must be performed for a specified drift rate which can be a function of *x* and *t* but cannot account for across-trial variability of drift rate. As a result, PyDDM does not offer native support for across-trial drift rate variability.

However, it is possible to approximate across-trial drift rate variability within the Fokker-Planck approach by discretizing the drift rate’s probability distribution, running a Fokker-Planck numerical solution for each drift rate separately, and then combining the results to yield an estimation of RT distributions. As the reviewer pointed out, this feature would be of interest to users.

In response to the reviewer’s concern, and to address this use case, we have now added an example to the PyDDM online documentation demonstrating how to implement across-trial drift rate variability:

https://pyddm.readthedocs.io/en/latest/cookbook/driftnoise.html#across-trial-variability-indrift-rate

We note that this support for across-trial drift rate variability is not without cost, due to limitations of the Fokker-Plank approach. In particular, this will cause simulations to be slow depending on the discretization of the drift rate distribution (due to repeated numerical solves of Fokker-Planck), and will interfere with the ability to automatically compute model likelihood. Thus, as the reviewer points out, it must be made clear to the reader that PyDDM does not have “first class” support for this feature, although it is implementable.

In summary, the original manuscript contained the following elements to highlight the fact that PyDDM does not support drift rate variability:

Mentioned it explicitly in the subsection “GDDM as a generalization of the DDM”.

– Highlighted this as a major negative for PyDDM (a “red cell”) in the table comparing software packages.

– Showed this in the diagram in Figure 1 by not drawing a distribution around the arrow (as we did for the “full DDM”).

– Included a paragraph in the Discussion section about it.

– Discussed technical aspects of not fitting the “full DDM” in the Materials and methods section.

Following the reviewer’s suggestion, we have changed our phrasing of “single-trial variability” to “across-trial variability” throughout the manuscript.

In addition to the elements listed above which were already in the manuscript, we made the following changes to further emphasize the fact that PyDDM does not natively support across-trial drift rate variability:

- Within the figure caption of Figure 1, we explicitly pointed out the meaning of the lack of a distribution around the arrow representing drift. The relevant portion of the caption now reads:

"The GDDM is a generalization of the DDM framework which allows arbitrary distributions for starting position and non-decision time, as well as arbitrary functions (instead of distributions) for drift rate and collapsing bounds."

- We change the text within the table to read "slow discretization (via extension)" under the “Across-trial drift variability” section

- We rewrote the paragraph in the Discussion section to be the following, to better explain our GDDM framework vis-`a-vis across-trial drift rate variability:"

The GDDM is a generalization of the DDM framework which allows arbitrary distributions for starting position and non-decision time, as well as arbitrary functions (instead of distributions) for drift rate and collapsing bounds."

- We change the text within the table to read "slow discretization (via extension)" under the “Across-trial drift variability” section

- We rewrote the paragraph in the Discussion section to be the following, to better explain our GDDM framework vis-`a-vis across-trial drift rate variability:

"The GDDM framework supports across-trial variability in non-decision time and starting position according to any arbitrary distribution, but it does not support across-trial drift rate variability. […] GDDMs offer researchers the capability to explore alternative cognitive mechanisms which might also account for slow errors."

2) The demonstration where the models are fit to the data is (deeply) problematic, because the monkey data are simply not representative. As can be seen from the grey histograms in Figure 2, the "regular" DDMs predict a long-tailed RT distribution, which is however not apparent from the data. This then creates ambiguity: yes, one can show that the existing models do not fit well, but they are fit to unrepresentative data. If these data patterns were regularly observed, the DDM would never have been proposed in the first place. It would be much more compelling to fit the models to any of the large benchmark data sets that are publicly available – these data sets all show a pronounced right skew.

We thank the reviewer for this important comment. Please see our response above to Essential revision point 2, which covers this point in depth.

3) As the progenitor of the much-maligned "EZ" model, I guess I am obliged to protest a little. Yes, of course EZ is not flexible, and does not have all the bells and whistles. But that is kind of the point of a simple model. There are some demonstrations that (contrary to my own expectations even) EZ actually does better than the full diffusion model when applied to sparse data. I am not talking about goodness-of-fit, but about recovering the ground truth in a simulation study (e.g., figuring out which parameters are the same and which differ). For sparse data, a simple model may work better. One reference:van Ravenzwaaij, Donkinand Vandekerckhove, (2017).

We thank the reviewer for pointing out the inadequacy of our discussion of the merits of EZ-Diffusion, especially its utility in the case of sparse data.

We have added the following passage to the Discussion section:

"Additionally, sometimes complex models which do represent appropriate cognitive mechanisms may exhibit inferior performance due to limitations of the available data. For instance, simulation studies suggest that when data are sparse, Bayesian fitting or EZ-Diffusion may be better able to recover parameters (Lerche et al., 2017; Wiecki et al., 2013; van Ravenzwaaij et al., 2017)."

Reviewer #3:Some concerns should be addressed before the manuscript can be published:1) The manuscript frequently describes the numerical solution of the FPE a "simulation", starting with the title, the introduction, the main text, etc. Commonly, solving the FPE is not considered a simulation, such that describing it as such is potentially confusing misnomer – in particular as there exist simulation-based "likelihood-free" methods to fit models solely based on simulating them. First, I urge the authors to not call solving FPEs a "simulation", but instead describe their approach as numerically propagating the probability mass, which can intuitively be thought of as performing an infinite number of simulations.

We agree with the reviewer that the term “simulation”, which usually refers to trial-wise trajectory simulations or sampling, is not the ideal terminology to use for Fokker-Planck. We agree that it is better to be technically precise with our language, to be clearer that we are not performing Monte Carlo simulations. As a result, throughout the main text we now use more precise language to describe the numerical Fokker-Planck solutions, describing propagation of the probability, rather than “simulation” which we use for Monte Carlo trajectories.

However, after careful consideration we have opted to keep the word “simulating” in the title, to accommodate the brevity and clarity of common usage required there. After considering different options, we believe using “simulating” in the title best communicates to a typical reader, in succinct and accessible language appropriate for a title, the generation of RT distributions for a specified GDDM. To the typical reader who would like to fit parameters of a GDDM to empirical data or estimate an RT distribution, it is not important whether it is through Monte Carlo trajectories or another method. We also note that usage of “simulation” of an RT distribution is arguably not technically incorrect, because the result of Fokker-Planck is the distribution of simulations as N → ∞. Likewise, by discretely sampling from the distribution produced by Fokker-Planck, it is indeed possible to use our framework for fast and efficient simulation of sets of RTs (albeit without originating trajectories of the decision variable).

Within the body text, we have replaced every instance of “simulate” with more descriptive and/or precise terminology when it refers to either Fokker-Planck, or collectively to both Fokker-Planck and trial-wise trajectory simulations. In most cases, this involved replacing “simulate” with “solve”. Overall, depending on the context, we made one of three substitutions: “solve”, when referring to Fokker-Planck or analytical solutions; “estimate the RT distribution”, when referring to the more general case which encompasses both Fokker-Planck and trial-wise trajectory simulation; and “post-factum modifications”, to replace “post-simulation modifications” (in reference to the “Overlay” object). As a result of these changes, the term “simulate” refers exclusively to trial-wise trajectory simulations in all but three places: the title, its first occurrence in the Abstract, and in one instance in the Introduction section where we specify the usage.

In order to ensure this does not cause confusion, we furthermore added the following to the Introduction section:

"Solving the Fokker-Planck equation is analogous to simulating an infinite number of individual trials."

Second, the authors should acknowledge likelihood-free fitting methods, like ABC (see, for example, Holmes and Trueblood, 2018 and related) to fit diffusion models – subsection “Methods for simulating the DDM and GDDM” would be a good starting point.

We have changed the suggested line as follows:

"Since this method does not produce a probability distribution, it is difficult to use efficient and robust methods such as full-distribution maximum likelihood for fitting data, though progress is ongoing in this area using likelihood-free methods such as through approximate Bayesian computation (Holmes and Trueblood, 2018)."

2) The precision and the clarity of the manuscript can be improved in many:

We thank the reviewer for these comments. We agree that because language in the manuscript has heavily emphasized readability for typical target users, mathematical precision may be under-emphasized in the main text at times.

- It should be made clear that GDDMs only support Markovian processes + some additional post-processing of the computed first-passage time distributions. This is, for example, the reason why they don't support continuously varying drift rates, but this isn't at all clear from the manuscript.

The reviewer is correct in that we do not support non-Markovian processes, for instance mechanisms related to momentum and inertia.

We have added the following text to clarify this point:

"The GDDM framework supports Markovian processes, in that the instantaneous drift rate and diffusion constant of a particle is determined by its position *xt* and the current time *t*, without dependence on the prior trajectory leading to that state. Drift rate is discretized, such that the timestep of the discretization is equal to the timestep of the Fokker-Planck solution. In practice, for typical task paradigms and dataset properties, the impact of this discretization is expected to be minimal for a reasonable timestep size."

An additional (current) limitation is that the implementation currently only supports boundaries that vary symmetrically around zero, even though this appears to be a limitation of the provided implementation rather than a fundamental limit to the chosen approach.

The reviewer is correct in that PyDDM currently only supports bounds which are symmetric around zero, that this is due to the implementation rather than the methodology, and that adding asymmetric bounds is a feasible future addition.

We have added the following text to clarify this point:

"Second, while PyDDM currently only implements bounds which are symmetric around zero, support for asymmetric bounds is a feasible future addition."

- The manuscript frequently claims that the solution is "continuous" (e.g., subsection “Methods for simulating the DDM and GDDM”, subsection “Software package for GDDM fitting”) in time, even though they only provide time-discretized numerical results.

Our use of the word “continuous” in the manuscript overwhelmingly refers to “continuous maximum likelihood”, a term borrowed from Heathcote, Brown and Mewhort, (2002). It is not sufficient to describe our fitting procedure as “maximum likelihood”, because this can refer to a number of fitting procedures, such as quantile maximum likelihood (an approximation to likelihood based on data quantiles; Heathcote, Brown and Mewhort, 2002), or to the maximum likelihood derivations which utilize only summary statistics of the RT distribution rather than all of the data points (e.g. Palmer et al., 2005).

However, we also see the reviewer’s point that our use of the term “continuous” could be confusing in this context.

We have replaced all references to “continuous maximum likelihood” with “full-distribution maximum likelihood”. We have also confirmed that no other uses of the word “continuous” refer to a discrete approximation.

- It isn't sufficiently clear from the manuscript that all provided FPE solutions are approximate, as they discretize time and space. Currently, the manuscript only highlights the approximate nature of the forward Euler scheme (subsection “Execution time vs error tradeoff”). Indeed, it has worse numerical properties, but that doesn't make the implicit schemes non-approximate.

We thank the reviewer for pointing out our asymmetric use of the term “approximation”.

We updated the referenced lines to say the following:

"…forward Euler iteratively approximates the probability distribution of trajectory position at each timestep using the distribution at the previous timestep, while backward Euler and Crank-Nicolson iteratively solve systems of linear equations to approximate the probability distribution (Voss and Voss, 2008)."

3) The manuscript highlights a better fit (i.e., higher likelihood, better BIC, etc.) of a DDM with time-varying boundaries and a leak, and uses this fit (of a single dataset) to re-iterate throughout the manuscript that a better model fit with fewer parameters is always more desirable (e.g., subsection “Software package for GDDM fitting”). This mis-characterizes the modeling exercise, in which the structure and presence of different components of DDMs might be as important as the quality of the fit.

We thank the reviewer for pointing out that our description of model fitting could be misleading. For a full description of how this point was addressed, see our response above to Essential revisions point 2.

4) The authors highlight three approaches to "simulate" DDMs and GDDMs, but seem to be missing a fourth based on solving Volterra integral equation, as described by Smith, (2000) and implemented by the 'dm' package by Drugowitsch et al., – which is fairly flexible, but unfortunately only provides computing the first-passage times, but no model fitting code.

We thank the reviewer for bringing to our attention that we have not properly discussed the method of Smith, (2000).

We have modified the passage describing the “three methods” for solving the GDDM such that the method of Smith, (2000) is incorporated into the third method. The text is now as follows:

"A third and better method is to iteratively propagate the distribution forward in time, which is achieved by solving the Fokker-Planck equation (Voss and Voss, 2008) or using the method of Smith, (2000). These methods allow the RT distribution of GDDMs to be estimated with better performance and lower error than trial-wise trajectory simulation. However, they are difficult to implement, which has thus far impeded their widespread use in cognitive psychology and neuroscience."

We also added the following passage to the Discussion section to bring attention to this fourth method and its relationship to PyDDM:

"PyDDM uses numerical solutions of the Fokker-Planck equation to estimate the RT distribution of the model. Other methods may be used for this estimation, such as solving the Volterra integral equation (Smith, 2000). Some prior work has used these alternative methods for solving GDDMs (Drugowitsch et al., 2012; Zhang et al., 2014; Drugowitsch et al., 2014). Such methods already have fast solvers (Drugowitsch, 2016), which could be implemented as an alternative method within PyDDM in the future."